

# Correlation functions by separation of variables: The XXX spin chain

**Giuliano Niccoli[1][⋆], Hao Pei[2] and Véronique Terras[2][†]**

**1** Univ Lyon, Ens de Lyon, Univ Claude Bernard, CNRS,
Laboratoire de Physique, F-69342 Lyon, France
**2** Université Paris-Saclay, CNRS, LPTMS, 91405, Orsay, France

⋆ giuliano.niccoli@ens-lyon.fr, † veronique.terras@universite-paris-saclay.fr

## Abstract

We explain how to compute correlation functions at zero temperature within the framework of the quantum version of the Separation of Variables (SoV) in the case of a simple model: the XXX Heisenberg chain of spin 1/2 with twisted (quasi-periodic) boundary conditions. We first detail all steps of our method in the case of anti-periodic boundary conditions. The model can be solved in the SoV framework by introducing inhomogeneity parameters. The action of local operators on the eigenstates are then naturally expressed in terms of multiple sums over these inhomogeneity parameters. We explain how to transform these sums over inhomogeneity parameters into multiple contour integrals. Evaluating these multiple integrals by the residues of the poles outside the integration contours, we rewrite this action as a sum involving the roots of the Baxter polynomial plus a contribution of the poles at infinity. We show that the contribution of the poles at infinity vanishes in the thermodynamic limit, and that we recover in this limit for the zero-temperature correlation functions the multiple integral representation that had been previously obtained through the study of the periodic case by Bethe Ansatz or through the study of the infinite volume model by the q-vertex operator approach. We finally show that the method can easily be generalized to the case of a more general non-diagonal twist: the corresponding weights of the different terms for the correlation functions in finite volume are then modified, but we recover in the thermodynamic limit the same multiple integral representation than in the periodic or anti-periodic case, hence proving the independence of the thermodynamic limit of the correlation functions with respect to the particular form of the boundary twist.

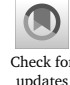

# 1  Introduction

In this paper we introduce an approach to compute the correlation functions of the quantum integrable lattice models that can be solved in the framework of the quantum Separation of Variables (SoV) method [1–6]. We develop here our approach in the case of a very simple model: the XXX Heisenberg spin-1/2 chain with quasi-periodic boundary conditions.

While outstanding successes have been achieved concerning the exact determination of the spectrum of quantum integrable systems, the exact computation of the correlation functions still remains a substantially more complicated problem. In fact, nowadays, exact results for correlation functions are available only for a very restricted set of quantum integrable models.

In the framework of the Quantum Inverse Scattering Method (QISM) and of the algebraic version of the Bethe Ansatz (ABA) [7–16], computations of zero-temperature correlation functions of some quantum integrable models, like the Heisenberg XXZ spin-1/2 chain with periodic boundary conditions, have been developed in [17–23]. Unlike previous methods based on the q-deformed KZ equations (the massless regime) and on the Baxter corner transfer matrix and q-vertex operator techniques (the massive regime) [24–28], the ABA approach can be directly applied to finite chains in a constant magnetic field. Note that the approach of [17–23] relies mainly on three essential ingredients: i. the expression of local operators in terms of the elements of the quantum monodromy matrix (solution of the quantum inverse problem) [17,18,29], which enables one to compute their action on the Bethe states by means of the Yang-Baxter commutation relations, ii. the use of a compact determinant representation for the scalar products of the so-called Bethe states in terms of the Bethe roots (Slavnov's scalar product formula) [30] and iii. a precise description of the configuration of Bethe roots for the ground state, given in the thermodynamic limit in terms of a density function solution of an integral equation on an interval of the real axis [31–33]. Further developments of this ABA approach also led to the numerical computation of dynamical structure factors [34]

(quantities that are directly accessible experimentally through neutron scattering [35]) and to the analytical asymptotic study at long distances of the two-point or multi-point functions in the thermodynamic limit [36–45]. Correlation functions can also be computed in the temperature case by the use of the so-called Quantum Transfer Matrix tools [46–51]. Let us also mention the existence of an alternative algebraic approach to correlation functions, in relation with a hidden Grassmann structure [52–60]. See also some subsequent papers on correlation functions [61, 62].

Let us however stress that these results have essentially been obtained for very simple models such as the XXZ spin chain or the quantum non-linear Schrödinger model with periodic boundary conditions. For more complicated integrable models or different type of boundary conditions, the situation may become much more cumbersome. On the one hand, it may happen that some physically interesting integrable models are not directly solvable by ABA, as for instance the open XXZ spin chain with general[1] boundary magnetic fields. In that case, other methods have to be used to construct the transfer matrix eigenstates, but they are still under development in what concerns the computation of correlation functions[2]. On the other hand, even for models for which standard ABA is in principle applicable and for which the spectrum and eigenstates are known, the generalization of one of the above outlined essential ingredients for the computation of correlation functions is often missing. In particular, the obtention of a generalization of the Slavnov's formula [30] for the scalar products of Bethe states, and more generally of a similar determinant representation for the matrix elements of local operators, as in [17], may be a very difficult problem if the combinatorial structure of the Bethe states is too involved. This is for instance the case in the XYZ model, for which first results about scalar products within ABA were obtained only very recently in [73] (but for which the obtention of a compact formula for matrix elements of local operators in finite volume remains an open problem[3]), using the fact that the scalar products of on-shell/off-shell Bethe vectors can be characterized as solutions to a system of linear equations, as initially proposed in [76]. This is also the case for models based on higher rank algebras, see for instance the works [77–85].

For quantum integrable models in the QISM framework, the limitation of the range of applicability of the ordinary ABA can be notably overcome by the use of SoV, which appears to have a much wider range of applicability. In fact, the latter approach has by now been systematically developed for rank one integrable quantum models [86–116] and more recently widely extended even to higher rank cases in [117–124], see also [6, 88, 125, 126] for previous developments. Moreover, the use of SoV has several other advantages, notably the fact that the completeness of the transfer matrix spectrum is a built-in feature. Another advantage with respect to the ABA approach concerns the fact that scalar products of *separate states*[4] can be generically expressed in the form of determinants, at least for rank one models[5], [100, 102–

---

[1]For $z$-oriented boundary magnetic fields, the model is solvable by ordinary Bethe Ansatz or by the $q$-vertex operator approach, and there exist exact representations for the correlation functions [27, 28, 63, 64].

[2]Here we are referring not only to the Separation of Variables — the subject of the present article — but also to an interesting modification of the Bethe Ansatz (the so-called *modified algebraic Bethe Ansatz*), introduced in [65–68] and developed further in [69–71] in what concerns the computations of scalar products of Bethe states, a first step towards correlation functions. Let us also mention in this context the so-called *off-diagonal Bethe Ansatz* which was proposed to describe the spectrum of models without U(1) symmetry [72] (the corresponding eigenstates being anyway constructed through SoV).

[3]Note however that some integral representations for the correlation functions could be obtained in this case in [74, 75] by means of the q-vertex operator approach directly in the infinite volume limit.

[4]A class of states with factorized wave-functions in the SoV basis, which notably includes the eigenstates of the transfer matrix.

[5]Determinant formulae for scalar products of separate states have also emerged recently in [127] for the higher rank gl(3) case, under special choices of the conserved charges generating the SoV bases. Let us also mention the interesting and recent papers [128, 129], also dealing with the computations of higher rank scalar products in a related SoV framework.

108, 111, 116]. Nevertheless, despite the impressive range of applicability of SoV, a general approach to correlation functions is so far missing within this approach. In fact, there are only very few results on correlation functions deduced by the use of SoV, see for example [86].

In fact, the main difficulties for the computation of physical quantities such as correlation functions in the SoV approach come from the fact that, for the method to apply, one has to deform the model by inhomogeneity parameters. The spectrum and eigenstates of the deformed model, as well as the determinant representations for the scalar products of separate states, are then characterized in terms of these inhomogeneity parameters. Coming back to the original physical model, i.e. having a description of the spectrum and a representation of the scalar products in which the homogeneous limit can be taken naturally, may not be an easy task. At the level of the spectrum, it usually means that one should transform the discrete SoV description into a more conventional one, for instance in terms of Q-functions solving TQ-equations of Baxter's type [130]. This may sometimes be simply done by polynomial interpolation, as in the case of the quasi-periodic XXX model [113, 117], but the situation may be more complicated if the resulting Q-function (i.e. the corresponding eigenvalues of the Q-operator) have no longer the same functional form as the usual functions of the model, as for instance in the anti-periodic XXZ case (see [131] for the construction of the Q-operator and [110] for a proof of the equivalence with the SoV description of the spectrum), in the quasi-periodic XYZ case (see [111,112]), or in the case of open chains with general boundary fields, where only incomplete results could be obtained so far [109,132]. Note that this difficulty may be overcome, as initially suggested in the context of the so-called off-diagonal Bethe Ansatz [72], by the consideration of TQ-equations with an additional term (called *inhomogenous* TQ-equations[6]) so that the Q-function is still a (usual, trigonometric, elliptic...) polynomial and that its equivalence with the SoV description of the spectrum can still easily be proven by mere polynomial interpolation (see for instance [109] for a proof of such a reformulation in the open XXZ chain with completely general boundary fields). This presents the clear advantage of providing a description of the spectrum in cases in which such a description in terms of a usual TQ-equation is still unknown, as in [109]. However, the disadvantage is that the presence of the inhomogeneous term complicates a priori drastically the analysis of the Bethe roots configurations, of their behavior in the thermodynamic limit and of the control of the finite-size corrections (see nevertheless [133] for a numerical investigation of these Bethe roots in a particular case for which a comparison with solutions of usual Bethe equations is possible). As for the scalar products, they could be transformed into determinants of Slavnov's type[7] in [113] in the XXX case (see also [115, 134] for open spin chains with some constraints on the boundary), but one should mention that the generalization of these transformations to the anti-periodic XXZ case is already not so obvious [135]. One should also mention that an explicit computation of the correlation functions as was done through other approaches in [17,24] implies several other non-trivial steps (computation of the multiple action of a product of local operators on eigenstates, analyzing the obtained formulas in the thermodynamic limit...) that have not been tackled so far within the SoV approach, even in a simple model such as the XXX spin chain.

This is the purpose of the present article to fill this gap: we explain here how to compute the correlation functions within the SoV approach, hence showing that it is possible to fully overcome the intrinsic difficulty of the approach related to the apparent omnipresence of the inhomogeneity parameters. Our method demands as pre-requirements the transfer matrix complete spectrum characterization (for instance in terms of Q-functions solving a Baxter TQ-

---

[6]Let us mention here that such inhomogeneous TQ-equations also appear naturally in the context of the modified algebraic Bethe Ansatz [65–71, 133].

[7]i.e. into determinants which have similar forms as the one obtained in [30], with in particular rows and columns labelled by the roots of the corresponding Q-function (and no longer by the inhomogeneity parameters as those obtained naturally by SoV).

equation), suitable determinant representations for the scalar products of the separate states, and the reconstruction of the local operators in the SoV representation. We develop here our method in the case of the XXX spin chain, but we expect it to be adaptable to other models for which the three aforementioned pre-requirements are fulfilled.

The main steps of our method can be summarized as follows. We first compute the action of products of local operators on the transfer matrix eigenstates by using their reconstruction in terms of the SoV representation. This results in multiple sums of separate states over the spectrum of the separate variables. The latter being expressed in terms of the inhomogeneity parameters of the model, we need to reformulate these multiple sums into a more convenient form. To this aim, we transform them into multiple contour integrals that we can evaluate by their residues at the poles outside the integration contours, as a sum involving the roots of the corresponding Baxter polynomial (the "Bethe roots") plus further possible contributions like poles at infinity. Hence the correlation functions at zero-temperature, or more precisely their elementary building blocks (i.e. the mean values of any product of local operators in the ground state) can be rewritten as a sum over scalar products of particular separate states. Using the determinant formula for these scalar products and the thermodynamic distribution of the ground state Bethe roots, we can analyze the thermodynamic behavior of each term of the sum, showing that many of them actually vanish in the thermodynamic limit. The non-vanishing terms can then be rewritten in the form of multiple integrals in this limit, as in [17, 24].

As already mentioned, we implement our approach here by considering one of the simplest models solvable by SoV: the XXX spin 1/2 chain with twisted (quasi-periodic) boundary conditions. For clarity, we choose to detail all steps of the methods in the specific case of anti-periodic boundary conditions, given by the twist matrix $\sigma^x$. In the last part of the paper, we explain how all these steps can be generalized in the case of a generic (non-diagonal) twist matrix $K$. We explicitly show that the thermodynamic limit of the zero-temperature correlation functions is invariant with respect to these quasi-periodic boundary conditions, i.e. with respect to the specific form of the twist matrix $K$, hence coinciding, in agreement with physical expectations, with the results obtained in the periodic case by Bethe Ansatz [17] or through the study of the infinite volume model by the q-vertex operator approach [24].

Let us stress here that these results are interesting in their own, and not only for the method that we have developed. As already mentioned, this provides an explicit derivation, from exact computations on the finite lattice, of the fact that the correlation functions in the thermodynamic limit do not depend on the boundary conditions that we impose — at least for quasi-periodic chains. Moreover, one has to point out that, contrary to what happens for the form factors of a single local operator [113], the elementary building blocks for the correlation functions that we have computed here cannot in general be simply deduced from the corresponding ABA results by using the GL(2) symmetry of the model. Indeed, taken a non-diagonal twist $K$ which is diagonalizable, then the GL(2) symmetry only implies that the transfer matrix associated to the non-diagonal twist is similar to that of the diagonal one. While this similarity relation allows one to compute the spectrum of one transfer matrix in terms of the other one, it does not lead to non-trivial relations between their elementary blocks. More precisely, an elementary block of size $m$ for the original transfer matrix with non-diagonal twist is transformed into a sum of up to $4^m$ elementary blocks for the similar transfer matrix with diagonal twist. Some of these elementary blocks can be shown to be zero on the basis of the symmetry of the diagonal model, but nevertheless in general one still need to consider a huge sum of elementary blocks if one pretends to use ABA methods, see appendix A.

The paper is organized as follows. After briefly introducing the anti-periodic XXX spin 1/2 chain in section 2, we recall the SoV solution of this model in section 3, and we more specifically describe the ground state of the model in section 4. In section 5, we explain how to com-

pute the correlation functions, or more precisely their elementary building blocks (or in other words the density matrix elements of a segment of length $m$), for the finite size chain. More precisely, we derive the multiple actions of local operators on the transfer matrix eigenstates, which enables us to express the correlation functions as multiple sums over scalar products of some separate states. We recall the explicit determinant representation for these scalar products. In section 6, we consider the thermodynamic limit of the previous multiple sums for the correlation functions in the ground state. We show that many terms of these sums vanish in the thermodynamic limit, and characterize the terms that remain finite in this limit. We hence recover, in this limit, the same selection rules as for the elementary building blocks of the periodic chain, and the same multiple integral representations for the non-vanishing terms. In section 7, we explain how all this procedure can be adapted to the case of a more general non-diagonal boundary twist $K$, and show that it produces the same result for the elementary building blocks of the correlation functions in the thermodynamic limit, hence proving the independence of these thermodynamic limit expressions with respect to the particular form of the boundary twist $K$. Finally, in appendix A, we make some comments about the transformation of the elementary building blocks for the correlation functions with respect to $GL(2)$ gauge transformations.

## 2    The anti-periodic XXX model

Let us consider the XXX Heisenberg chain of spin 1/2,

$$H = \sum_{n=1}^{N} \left[ \sigma_n^x \sigma_{n+1}^x + \sigma_n^y \sigma_{n+1}^y + \sigma_n^z \sigma_{n+1}^z - 1 \right]. \tag{1}$$

Here and in the following, $\sigma_n^a$, $a = x, y, z$, stand for the Pauli matrices at site $n$, acting on the local quantum spin space $V_n \simeq \mathbb{C}^2$. We moreover impose twisted boundary conditions. For simplicity, we shall mainly focus, until section 6, on the case of anti-periodic boundary conditions with twist matrix $\sigma^x$,

$$\sigma_{N+1}^a = \sigma_1^x \sigma_1^a \sigma_1^x, \qquad a = x, y, z, \tag{2}$$

but in section 7 we shall also extend our study to the case of a more general twist matrix $K$.

The monodromy matrix of the inhomogeneous version of the XXX spin-1/2 chain is defined as

$$T_0(\lambda) = R_{0N}(\lambda - \xi_N) \ldots R_{01}(\lambda - \xi_1) = \begin{pmatrix} A(\lambda) & B(\lambda) \\ C(\lambda) & D(\lambda) \end{pmatrix}_{[0]}, \tag{3}$$

where $\lambda$ is the so-called spectral parameters, $\xi_1, \ldots, \xi_N$ are inhomogeneity parameters, and where $R(\lambda)$ is the $R$-matrix of the model. The latter is of the form

$$R(\lambda) = \begin{pmatrix} \lambda + \eta & 0 & 0 & 0 \\ 0 & \lambda & \eta & 0 \\ 0 & \eta & \lambda & 0 \\ 0 & 0 & 0 & \lambda + \eta \end{pmatrix}, \tag{4}$$

where $\eta$ is an arbitrary non-zero complex parameter. The transfer matrix of the model with anti-periodic boundary conditions is

$$\mathcal{T}(\lambda) = \text{tr}_0 \left[ \sigma_0^x T_0(\lambda) \right] = B(\lambda) + C(\lambda). \tag{5}$$

It is a polynomial in $\lambda$ of degree $N-1$, which moreover satisfies the symmetries

$$[S^x, \mathcal{T}(\lambda)] = 0, \qquad S^x = \sum_{n=1}^{N} \sigma_n^x, \tag{6}$$

$$[\Gamma^x, \mathcal{T}(\lambda)] = 0 \qquad \Gamma^x = \overset{N}{\underset{n=1}{\otimes}} \sigma_n^x = (-i)^N \exp\left[\frac{i\pi}{2} S^x\right]. \tag{7}$$

In the homogeneous limit $\xi_n \to \eta/2$, $n = 1, \ldots, N$, the Hamiltonian (1) of the XXX spin $1/2$ chain with anti-periodic boundary conditions is recovered in terms of a logarithmic derivative of the anti-periodic transfer matrix (5) as

$$H = 2\eta\, \mathcal{T}(\lambda)^{-1} \frac{d}{d\lambda} \mathcal{T}(\lambda)\Big|_{\lambda=\eta/2} - 2N. \tag{8}$$

The quantum determinant, which is a central element of the Yang-Baxter algebra, can be expressed as

$$\det_q T(\lambda) = a(\lambda)\, d(\lambda - \eta) = A(\lambda)\, D(\lambda - \eta) - B(\lambda)\, C(\lambda - \eta)$$
$$= D(\lambda)\, A(\lambda - \eta) - C(\lambda)\, B(\lambda - \eta), \tag{9}$$

with

$$a(\lambda) = \prod_{n=1}^{N}(\lambda - \xi_n + \eta), \qquad d(\lambda) = \prod_{n=1}^{N}(\lambda - \xi_n). \tag{10}$$

## 3 Diagonalization of the transfer matrix by separation of variables

The diagonalization of the anti-periodic transfer matrix (5) was performed in [3,4] by separation of variables. Here we briefly recall the main results of this construction (see also [113]).

Let us suppose that the inhomogeneity parameters $\xi_1, \ldots, \xi_N$ are generic, or at least that they satisfy the condition

$$\xi_a \neq \xi_b \pm h\eta \quad \text{for} \quad h \in \{0, 1\}, \qquad \forall a \neq b. \tag{11}$$

Then, there exist a basis[8] $\{|\mathbf{h}\rangle, \mathbf{h} = (h_1, \ldots, h_N) \in \{0, 1\}^N\}$ of $\mathcal{H}$ and a basis $\{\langle \mathbf{h}|, \mathbf{h} = (h_1, \ldots, h_N) \in \{0, 1\}^N\}$ of $\mathcal{H}^*$ such that

$$D(\lambda)|\mathbf{h}\rangle = d_{\mathbf{h}}(\lambda)|\mathbf{h}\rangle = \prod_{n=1}^{N}(\lambda - \xi_n^{(h_n)})|\mathbf{h}\rangle, \tag{12}$$

$$C(\lambda)|\mathbf{h}\rangle = \sum_{a=1}^{N} \delta_{h_a,1}\, d(\xi_a^{(1)}) \prod_{b \neq a} \frac{\lambda - \xi_b^{(h_b)}}{\xi_a^{(h_a)} - \xi_b^{(h_b)}} |T_a^- \mathbf{h}\rangle, \tag{13}$$

$$B(\lambda)|\mathbf{h}\rangle = -\sum_{a=1}^{N} \delta_{h_a,0}\, a(\xi_a^{(0)}) \prod_{b \neq a} \frac{\lambda - \xi_b^{(h_b)}}{\xi_a^{(h_a)} - \xi_b^{(h_b)}} |T_a^+ \mathbf{h}\rangle, \tag{14}$$

---

[8]The explicit form of the SoV basis does not play any role in the computation of the correlation functions; so, we omit it and we refer to [113] for its explicit form.

and

$$\langle \mathbf{h} | D(\lambda) = d_{\mathbf{h}}(\lambda) \langle \mathbf{h} | = \prod_{n=1}^{N} (\lambda - \xi_n^{(h_n)}) \langle \mathbf{h} |, \tag{15}$$

$$\langle \mathbf{h} | C(\lambda) = \sum_{a=1}^{N} \delta_{h_a,0} \, d(\xi_a^{(1)}) \prod_{b \neq a} \frac{\lambda - \xi_b^{(h_b)}}{\xi_a^{(h_a)} - \xi_b^{(h_b)}} \langle \mathrm{T}_a^+ \mathbf{h} |, \tag{16}$$

$$\langle \mathbf{h} | B(\lambda) = -\sum_{a=1}^{N} \delta_{h_a,1} \, a(\xi_a^{(0)}) \prod_{b \neq a} \frac{\lambda - \xi_b^{(h_b)}}{\xi_a^{(h_a)} - \xi_b^{(h_b)}} \langle \mathrm{T}_a^- \mathbf{h} |. \tag{17}$$

Here we have set

$$\xi_n^{(h_n)} = \xi_n - h_n \eta \qquad \text{for} \quad h_n \in \{0,1\}, \tag{18}$$

$$d_{\mathbf{h}}(\lambda) = \prod_{n=1}^{N} (\lambda - \xi_n^{(h_n)}), \tag{19}$$

and

$$\mathrm{T}_a^{\pm}(h_1, \dots, h_N) = (h_1, \dots, h_a \pm 1, \dots, h_N). \tag{20}$$

To determine the action of $A(\lambda)$ on $|\mathbf{h}\rangle$ and on $\langle \mathbf{h}|$, one can use the quantum determinant relation (9). By using the first line of (9) and (12)-(14) we obtain:

$$
\begin{aligned}
A(\lambda) | \mathbf{h} \rangle &= \frac{\det_q T(\lambda) + B(\lambda) C(\lambda - \eta)}{d_{\mathbf{h}}(\lambda - \eta)} | \mathbf{h} \rangle \\
&= \frac{\det_q T(\lambda)}{d_{\mathbf{h}}(\lambda - \eta)} | \mathbf{h} \rangle + \frac{B(\lambda)}{d_{\mathbf{h}}(\lambda - \eta)} \sum_{a=1}^{N} \delta_{h_a,1} \, d(\xi_a^{(1)}) \prod_{\ell \neq a} \frac{\lambda - \eta - \xi_\ell^{(h_\ell)}}{\xi_a^{(h_a)} - \xi_\ell^{(h_\ell)}} | \mathrm{T}_a^- \mathbf{h} \rangle \\
&= \frac{\det_q T(\lambda)}{d_{\mathbf{h}}(\lambda - \eta)} | \mathbf{h} \rangle - \frac{1}{d_{\mathbf{h}}(\lambda - \eta)} \sum_{a=1}^{N} \delta_{h_a,1} \, d(\xi_a^{(1)}) \prod_{\ell \neq a} \frac{\lambda - \eta - \xi_\ell^{(h_\ell)}}{\xi_a^{(1)} - \xi_\ell^{(h_\ell)}} \\
&\quad \times \sum_{b=1}^{N} \delta_{(\mathrm{T}_a^- \mathbf{h})_b,0} \, a(\xi_b^{(0)}) \prod_{\ell \neq b} \frac{\lambda - \xi_\ell^{((\mathrm{T}_a^- \mathbf{h})_\ell)}}{\xi_b^{(0)} - \xi_\ell^{((\mathrm{T}_a^- \mathbf{h})_\ell)}} | \mathrm{T}_b^+ \mathrm{T}_a^- \mathbf{h} \rangle,
\end{aligned}
\tag{21}
$$

and by using the second line of (9) and (15)-(17):

$$
\begin{aligned}
\langle \mathbf{h} | A(\lambda) &= \langle \mathbf{h} | \frac{\det_q T(\lambda + \eta) + C(\lambda + \eta) B(\lambda)}{d_{\mathbf{h}}(\lambda + \eta)} \\
&= \langle \mathbf{h} | \frac{\det_q T(\lambda + \eta)}{d_{\mathbf{h}}(\lambda + \eta)} + \sum_{a=1}^{N} \delta_{h_a,0} \, d(\xi_a^{(1)}) \prod_{\ell \neq a} \frac{\lambda + \eta - \xi_\ell^{(h_\ell)}}{\xi_a^{(h_a)} - \xi_\ell^{(h_\ell)}} \langle \mathrm{T}_a^+ \mathbf{h} | \frac{B(\lambda)}{d_{\mathbf{h}}(\lambda + \eta)} \\
&= \langle \mathbf{h} | \frac{\det_q T(\lambda + \eta)}{d_{\mathbf{h}}(\lambda + \eta)} - \frac{1}{d_{\mathbf{h}}(\lambda + \eta)} \sum_{a=1}^{N} \delta_{h_a,0} \, d(\xi_a^{(1)}) \prod_{\ell \neq a} \frac{\lambda + \eta - \xi_\ell^{(h_\ell)}}{\xi_a^{(0)} - \xi_\ell^{(h_\ell)}} \\
&\quad \times \sum_{b=1}^{N} \delta_{(\mathrm{T}_a^+ \mathbf{h})_b,1} \, a(\xi_b^{(0)}) \prod_{\ell \neq b} \frac{\lambda - \xi_\ell^{((\mathrm{T}_a^+ \mathbf{h})_\ell)}}{\xi_b^{(1)} - \xi_\ell^{((\mathrm{T}_a^+ \mathbf{h})_\ell)}} \langle \mathrm{T}_b^- \mathrm{T}_a^+ \mathbf{h} |.
\end{aligned}
\tag{22}
$$

We have

$$\langle \mathbf{h} | \mathbf{k} \rangle = \frac{\delta_{\mathbf{h},\mathbf{k}}}{V(\xi_1^{(h_1)}, \dots, \xi_N^{(h_N)})}, \tag{23}$$

where, for any $n$-tuple $(x_1, \ldots, x_n)$, $V(x_1, \ldots, x_n)$ denotes the Vandermonde determinant

$$V(x_1, \ldots, x_n) = \prod_{\substack{i,j=1 \\ i<j}}^{n} (x_j - x_i). \tag{24}$$

The eigenvalues $\tau(\lambda)$ of the transfer matrix (5) are characterized by the fact that they are entire functions of $\lambda$ which can be written in the form

$$\tau(\lambda) = \frac{-a(\lambda)Q(\lambda - \eta) + d(\lambda)Q(\lambda + \eta)}{Q(\lambda)}, \tag{25}$$

in terms of a polynomial $Q(\lambda)$ of the form

$$Q(\lambda) = \prod_{j=1}^{R} (\lambda - \lambda_j), \qquad R \leq N, \tag{26}$$

for some set of roots $\lambda_1, \ldots, \lambda_R$ such that $\lambda_a \neq \xi_b$, $\forall a \in \{1, \ldots, R\}$, $\forall b \in \{1, \ldots, N\}$. For a given eigenvalue $\tau(\lambda)$ of the transfer matrix, the polynomial $Q$ satisfying these conditions is unique, and will therefore sometimes be denoted by $Q_\tau$. The corresponding left and right eigenstates of (5) with eigenvalue $\tau(\lambda)$ are obtained in terms of $Q_\tau$ as the states of the form

$$\langle Q_\tau | = \sum_{\mathbf{h} \in \{0,1\}^N} \prod_{n=1}^{N} Q_\tau(\xi_n^{(h_n)}) \, V(\xi_1^{(h_1)}, \ldots, \xi_N^{(h_N)}) \langle \mathbf{h} |, \tag{27}$$

$$|Q_\tau\rangle = \sum_{\mathbf{h} \in \{0,1\}^N} \prod_{n=1}^{N} \left\{ \left( -\frac{a(\xi_n)}{d(\xi_n - \eta)} \right)^{h_n} Q_\tau(\xi_n^{(h_n)}) \right\} V(\xi_1^{(h_1)}, \ldots, \xi_N^{(h_N)}) |\mathbf{h}\rangle$$

$$= \sum_{\mathbf{h} \in \{0,1\}^N} \prod_{n=1}^{N} Q_\tau(\xi_n^{(h_n)}) \, V(\xi_1^{(1-h_1)}, \ldots, \xi_N^{(1-h_N)}) |\mathbf{h}\rangle. \tag{28}$$

Hence, the eigenvalues and eigenstates of the anti-periodic transfer matrix can be characterized in terms of the (admissible) solutions of the Bethe equations for the roots $\lambda_1, \ldots, \lambda_R$ of $Q(\lambda)$, imposing that the quantity (25) is entire:

$$\mathfrak{a}_Q(\lambda_j) = 1, \qquad j = 1, \ldots, R, \tag{29}$$

where

$$\mathfrak{a}_Q(\lambda) = \frac{d(\lambda)}{a(\lambda)} \frac{Q(\lambda + \eta)}{Q(\lambda - \eta)}. \tag{30}$$

Moreover, the eigenstates (27)-(28) can be written in the form of generalized Bethe states[9] as

$$\langle Q_\tau | = (-1)^{RN} \langle 1 | \prod_{k=1}^{R} D(\lambda_k), \tag{31}$$

$$|Q_\tau\rangle = (-1)^{RN} \prod_{k=1}^{R} D(\lambda_k) |1\rangle, \tag{32}$$

---

[9]i.e. states that have a similar form as the Bethe states obtained though ABA, in that they are given by the multiple action, on some particular state (hence playing the role of a reference state), of some operator entry of the monodromy matrix evaluated at the Bethe roots.

where

$$\langle 1 | = \sum_{\mathbf{h} \in \{0,1\}^N} V(\xi_1^{(h_1)}, \ldots, \xi_N^{(h_N)}) \langle \mathbf{h} |, \tag{33}$$

$$| 1 \rangle = \sum_{\mathbf{h} \in \{0,1\}^N} V(\xi_1^{(1-h_1)}, \ldots, \xi_N^{(1-h_N)}) | \mathbf{h} \rangle, \tag{34}$$

are eigenvectors of the transfer matrix (5) with eigenvalue $-a(\lambda) + d(\lambda)$. Note that the eigenstates (31)-(32) can alternatively be written in the form

$$\langle Q_\tau | = (-1)^N \frac{\prod_{k=1}^R d(\lambda_k)}{\prod_{k=1}^{N-R} d(\widehat{\lambda}_k)} \sum_{\mathbf{h}} \prod_{n=1}^N \left[ (-1)^{h_n} \widehat{Q}(\xi_n^{(h_n)}) \right] V(\xi_1^{(h_1)}, \ldots, \xi_N^{(h_N)}) \langle \mathbf{h} | \tag{35}$$

$$= (-1)^{RN} \frac{\prod_{k=1}^R d(\lambda_k)}{\prod_{k=1}^{N-R} d(\widehat{\lambda}_k)} \langle 1_{\mathrm{alt}} | \prod_{k=1}^{N-R} D(\widehat{\lambda}_k), \tag{36}$$

$$| Q_\tau \rangle = (-1)^N \frac{\prod_{k=1}^R d(\lambda_k)}{\prod_{k=1}^{N-R} d(\widehat{\lambda}_k)} \sum_{\mathbf{h}} \prod_{n=1}^N \left[ (-1)^{h_n} \widehat{Q}(\xi_n^{(h_n)}) \right] V(\xi_1^{(1-h_1)}, \ldots, \xi_N^{(1-h_N)}) | \mathbf{h} \rangle \tag{37}$$

$$= (-1)^{RN} \frac{\prod_{k=1}^R d(\lambda_k)}{\prod_{k=1}^{N-R} d(\widehat{\lambda}_k)} \prod_{k=1}^{N-R} D(\widehat{\lambda}_k) | 1_{\mathrm{alt}} \rangle, \tag{38}$$

where

$$\langle 1_{\mathrm{alt}} | = \sum_{\mathbf{h} \in \{0,1\}^N} \prod_{n=1}^N (-1)^{h_n} V(\xi_1^{(h_1)}, \ldots, \xi_N^{(h_N)}) \langle \mathbf{h} |, \tag{39}$$

$$| 1_{\mathrm{alt}} \rangle = \sum_{\mathbf{h} \in \{0,1\}^N} \prod_{n=1}^N (-1)^{h_n} V(\xi_1^{(1-h_1)}, \ldots, \xi_N^{(1-h_N)}) | \mathbf{h} \rangle, \tag{40}$$

are eigenvectors of (5) with eigenvalue $a(\lambda) - d(\lambda)$, and where

$$\widehat{Q}(\lambda) \equiv \widehat{Q}_\tau(\lambda) = \prod_{j=1}^{N-R} (\lambda - \widehat{\lambda}_j), \tag{41}$$

is the unique (up to normalization) polynomial solution with degree no more than $N$ of the TQ-equation with opposite signs:

$$\tau(\lambda) \widehat{Q}(\lambda) = a(\lambda) \widehat{Q}(\lambda - \eta) - d(\lambda) \widehat{Q}(\lambda + \eta). \tag{42}$$

Equivalently, $\widehat{Q}_\tau(\lambda) = Q_{-\tau}(\lambda)$ can be seen as the solution of (25) associated with the eigenvalue $-\tau(\lambda)$ of the transfer matrix, or $e^{i \frac{\pi}{\eta} \lambda} \widehat{Q}_\tau(\lambda)$ can be seen as the second (independent) solution of the TQ-equation (25) associated with the eigenvalue $\tau(\lambda)$. The two polynomials $Q(\lambda) \equiv Q_\tau(\lambda)$ and $\widehat{Q}(\lambda) \equiv \widehat{Q}_\tau(\lambda) = Q_{-\tau}(\lambda)$ satisfy the quantum wronskian relation:

$$\hat{W}_{Q, \widehat{Q}}(\lambda) = d(\lambda), \tag{43}$$

where

$$\hat{W}_{Q,\hat{Q}}(\lambda) = \frac{1}{2}\left[Q(\lambda)\hat{Q}(\lambda-\eta) + \hat{Q}(\lambda)Q(\lambda-\eta)\right]. \tag{44}$$

This means in particular that, if $Q(\lambda) \equiv Q_\tau(\lambda)$ has degree $R$, then $\hat{Q}(\lambda) \equiv \hat{Q}_\tau(\lambda) = Q_{-\tau}(\lambda)$ has indeed degree $N-R$.

Note that the expressions (25), (26), (29), (31)-(32) and (36), (38), (41), (42) are now suitable for the consideration of the homogeneous limit $\xi_1,\ldots,\xi_N \to \eta/2$ (provided that the homogeneous limit of the states $\langle 1|, |1\rangle$ and $\langle 1_{alt}|, |1_{alt}\rangle$ is well defined). In this limit, one recovers the physical model (1) and the states $\langle\Psi_\tau|$ (31), (36) and $|\Psi_\tau\rangle$ (32), (38) are eigenstates of the Hamiltonian with eigenvalue $E_\tau$ which can be expressed either in terms of the roots of $Q_\tau$ or of the roots of $\hat{Q}_\tau$:

$$E_\tau = \sum_{a=1}^{R} \frac{2\eta^2}{(\lambda_a - \eta/2)(\lambda_a + \eta/2)} = \sum_{a=1}^{N-R} \frac{2\eta^2}{(\hat{\lambda}_a - \eta/2)(\hat{\lambda}_a + \eta/2)}. \tag{45}$$

*Remark* 1. Since, if $\tau(\lambda)$ is an eigenvalue of the transfer matrix $\mathcal{T}(\lambda)$, $-\tau(\lambda)$ is also an eigenvalue (which is different from the previous one[10]), the spectrum of the Hamiltonian (1) obtained from (8) is doubly degenerated, with energy given in terms of the roots of $Q_\tau(\lambda)$ or of $\hat{Q}_\tau(\lambda) = Q_{-\tau}(\lambda)$ as in (45).

*Remark* 2. From the quantum wronskian relation (43)-(44), one can derive several relations between the roots $\lambda_j$, $j = 1,\ldots,R$ of $Q(\lambda) \equiv Q_\tau(\lambda)$ and the roots $\hat{\lambda}_j$, $j = 1,\ldots,N-R$ of $\hat{Q}(\lambda) \equiv \hat{Q}_\tau(\lambda) = Q_{-\tau}(\lambda)$. In particular, we have the sum rule:

$$\sum_{n=1}^{N}(\xi_n - \eta/2) = \sum_{j=1}^{R}\lambda_j + \sum_{j=1}^{N-R}\hat{\lambda}_j. \tag{46}$$

*Remark* 3. The eigenstates $|Q_\tau\rangle$ of the anti-periodic transfer matrix are also eigenstates of the symmetry operators $S^x$ (6) and $\Gamma^x$ (7):

$$S^x|Q_\tau\rangle = (N-2R)|Q_\tau\rangle, \qquad \Gamma^x|Q_\tau\rangle = (-1)^R|Q_\tau\rangle. \tag{47}$$

## 4 Description of the ground state

Let us now discuss the description of the ground state of the anti-periodic XXX chain (1) in terms of the solution of the Bethe equations (29).

We now consider the homogeneous limit $\xi_1,\ldots,\xi_N \to \eta/2$, and we set for convenience $\eta = -i$. The Bethe equations (29) then take the form

$$\left(\frac{i/2 - \lambda_j}{i/2 + \lambda_j}\right)^N \prod_{k=1}^{R} \frac{i + \lambda_j - \lambda_k}{i - \lambda_j + \lambda_k} = (-1)^{N-R}, \qquad j = 1,\ldots,R, \tag{48}$$

and the energy (45) associated with a configuration of Bethe roots $\{\lambda_j\}_{1\le j\le R}$ is

$$E(\{\lambda_j\}_{1\le j\le R}) = \sum_{a=1}^{R}\epsilon(\lambda_a), \qquad \text{with} \quad \epsilon(\lambda) = -\frac{2}{\lambda^2 + 1/4}. \tag{49}$$

We can show similarly as in [136] that the complex roots appear by pairs $z, \bar{z}$ for a solution with much more real roots than complex roots[11].

---

[10]Note that $\tau(\lambda)$ cannot be identically zero (even in the homogeneous limit) due to the fact that it satisfies the relations $\tau(\xi_n)\tau(\xi_n - \eta) = -a(\xi_n)d(\xi_n - \eta) \neq 0$.

[11]i.e. where the number of real roots is more than twice the number of complex roots.

For real roots $\lambda_j$, it is convenient, as in the periodic case, to rewrite the Bethe equations (48) in logarithmic form:

$$\widehat{\xi}_Q(\lambda_j) = \frac{2n_j - N + R}{N}\pi, \qquad n_j \in \mathbb{Z}, \tag{50}$$

where $\widehat{\xi}_Q(\lambda)$ is the counting function associated with a configuration of Bethe roots $Q$,

$$\widehat{\xi}_Q(\lambda) = \frac{i}{N}\log\left((-1)^{N-R}\mathfrak{a}_Q(\lambda)\right) = p(\lambda) + \frac{1}{N}\sum_{k=1}^{R}\theta(\lambda - \lambda_k), \tag{51}$$

with

$$p(\lambda) = i\log\left(\frac{i/2 + \lambda}{i/2 - \lambda}\right), \qquad p'(\lambda) = \frac{1}{\lambda^2 + 1/4} \tag{52}$$

$$\theta(\lambda) = i\log\left(\frac{i - \lambda}{i + \lambda}\right), \qquad \theta'(\lambda) = -\frac{2}{\lambda^2 + 1}. \tag{53}$$

Note that these Bethe equations are completely similar in their form to the ones that we have in the periodic case, the only difference being in the sign in the right hand side of (48). Hence the analysis of the solution is similar, except that this difference of sign will result in a difference in the allowed set of quantum numbers in the right hand side of (50).

*Remark* 4. We have however a crucial difference here with the periodic case: the SoV approach gives us the completeness of the corresponding Bethe states (at least if we slightly deform the model by inhomogeneity parameters), contrary to the periodic case for which Bethe states gives only $\mathfrak{su}(2)$ highest weight vectors. Moreover, we need here a priori to consider all degrees $R \le N$ of $Q$, and not only $R \le \frac{N}{2}$ as in the periodic case. Let us nevertheless remark that we can in fact avoid considering solutions of the Bethe equations "beyond the equator" (i.e. with $R > \frac{N}{2}$): we can indeed choose to construct the eigenstates associated with polynomials $Q$ with degree $R > \frac{N}{2}$ by (36)-(38), i.e. by means of the polynomial $\widehat{Q}$ which in that case has degree $N - R < \frac{N}{2}$.

As in the periodic case, we expect that, in the large $N$ limit, the low-energy states will be given by solutions $\{\lambda\} \equiv \{\lambda_1, \ldots, \lambda_R\}$ of the Bethe equations with an infinite number of real roots (of order $N/2$) and a finite number of complex roots. Let us also suppose that, for such states, the real Bethe roots have a continuous distribution in the thermodynamic limit:

$$\frac{1}{N(\lambda_{j+1} - \lambda_j)} \underset{N\to\infty}{\sim} \rho(\lambda_j), \qquad \text{if } \lambda_j, \lambda_{j+1} \in \mathbb{R}, \tag{54}$$

so that we suppose we can, in the leading order in the thermodynamic limit, replace the sums by integrals (see [33] for a proof in the periodic case):

$$\frac{1}{N}\sum_{k=1}^{R}f(\lambda_k) \underset{N\to+\infty}{\longrightarrow} \int_{-\infty}^{\infty} f(\lambda)\rho(\lambda)\,d\lambda, \tag{55}$$

for any sufficiently regular function $f$. The function $\rho(\lambda)$ is therefore solution of the integral equation

$$2\pi\rho(\lambda) - \int_{-\infty}^{\infty}\theta'(\lambda - \mu)\rho(\mu)\,d\mu = p'(\lambda), \tag{56}$$

which is the same integral equation as in the periodic case and therefore admits the same solution:

$$\rho(\lambda) = \frac{1}{2\cosh(\pi\lambda)}. \tag{57}$$

Note that we have

$$\widehat{\xi}'_Q(\lambda) = \frac{i}{N} \frac{\mathfrak{a}'_Q(\lambda)}{\mathfrak{a}_Q(\lambda)} \xrightarrow[N\to\infty]{} 2\pi\rho(\lambda). \tag{58}$$

The function $p$ (resp. $\theta$) is holomorphic in a band of width $i$ (resp. $2i$) around the real axis. $p$ and $\theta$ (and hence $\widehat{\xi}$) are odd functions of $\lambda$. Moreover,

$$p(\lambda) \xrightarrow[\Re(\lambda)\to\pm\infty]{} \pm\pi, \qquad \text{if} \quad |\Im(\lambda)| < \frac{1}{2}, \tag{59}$$

$$\theta(\lambda) \xrightarrow[\Re(\lambda)\to\pm\infty]{} \mp\pi, \qquad \text{if} \quad |\Im(\lambda)| < 1, \tag{60}$$

so that, if all roots are *close roots* (i.e. such that $|\Im(\lambda_k)| < 1$, $k = 1, \ldots, R$),

$$\widehat{\xi}(\lambda) \xrightarrow[\lambda\to\pm\infty]{} \pm\frac{N-R}{N}\pi, \qquad \text{for} \quad \lambda \in \mathbb{R}. \tag{61}$$

Hence, if we suppose that the counting function is an increasing function and if all roots are close roots, the allowed set of quantum numbers $n_j$ in (50) would be

$$n_j \in \{1, \ldots, N-R-1\}, \tag{62}$$

which means in particular that we could have at most $N-R-1$ real Bethe roots in a sector with $R$ Bethe roots.

The question is whether the counting function is indeed an increasing function. This should be true on any compact interval of the real axis and for $N$ large enough due to (58). However, nothing assures us it is true on the whole real axis, which is non-compact. To clarify this point, let us evaluate the derivative of the counting function at large values of $\pm\lambda$:

$$\begin{aligned}
\widehat{\xi}'(\lambda) &= \frac{1}{1+1/4} + \frac{1}{N}\sum_{k=1}^{R} \frac{1}{(\lambda-\lambda_k)^2 + 1} \\
&= \frac{N-2R}{N\lambda^2} - \frac{4}{N\lambda^3}\sum_{k=1}^{R}\lambda_k + O(1/\lambda^4).
\end{aligned} \tag{63}$$

Hence, if $N-2R > 0$, the counting function is indeed strictly increasing at large $\lambda$. This does not prove that it is increasing on the whole real axis but at least it does not contradict this hypothesis.

On the contrary, if $N-2R < 0$, the counting function is strictly decreasing at large $\lambda$. This means that the restriction (62) is certainly not valid in that case, since both limiting values in (61) can in fact be reached for finite values of $\lambda$ and therefore should be included in the set of allowed integers. Hence, we have (at least) $N-R+1$ possible vacancies on the real axis in that case.

In the particular case $N = 2R$ for $N$ even, the sign of $\widehat{\xi}'(\lambda)$ is given by the sign of the sum of Bethe roots:

$$\widehat{\xi}'(\lambda) \begin{cases} < 0 \text{ if } \sum\lambda_k > 0 \\ > 0 \text{ if } \sum\lambda_k < 0 \end{cases} \qquad \text{when } \lambda \to +\infty, \tag{64}$$

$$\widehat{\xi}'(\lambda) \begin{cases} > 0 \text{ if } \sum\lambda_k > 0 \\ < 0 \text{ if } \sum\lambda_k < 0 \end{cases} \qquad \text{when } \lambda \to -\infty. \tag{65}$$

Hence, in that case (provided that $\sum\lambda_k \neq 0$), one of the limiting value in (61) can be reached for finite $\lambda$. It means that we have (at least) $N/2$ possible vacancies on the real axis. It is

therefore natural to expect that, for $N$ even, the ground state of the model is given by a state with exactly $R = N/2$ real roots, as in the periodic and the diagonal twist cases[12]. Note that, from Remark 1, the ground state is doubly degenerated. We have indeed two such states related to $Q$ and $\widehat{Q}$ with the same numbers of roots $\lambda_1, \ldots, \lambda_{N/2}$ and $\widehat{\lambda}_1, \ldots, \widehat{\lambda}_{N/2}$, and the sum rule (46) imposes moreover that

$$\sum_{k=1}^{N/2} \lambda_k = -\sum_{k=1}^{N/2} \widehat{\lambda}_k, \tag{66}$$

in the homogeneous limit. Hence we expect these two states to have adjacent sets of quantum numbers shifted by one with respect to each other.

For $N$ odd, instead, we expect that the two degenerate ground states are in the two different sectors $R = \frac{N-1}{2}$ and $R = \frac{N+1}{2}$. In the sector $R = \frac{N-1}{2}$, there are indeed from our previous study (at least) $\frac{N-1}{2}$ possible vacancies on the real axis. Hence, there exists a solution in that sector with only real roots $\lambda_1, \ldots, \lambda_{\frac{N-1}{2}}$ which should be the ground state. In the sector $R = \frac{N+1}{2}$, we have a state with the same energy, which correspond to a polynomial $\widehat{Q}$ with $N - R = \frac{N-1}{2}$ real roots $\widehat{\lambda}_1, \ldots, \widehat{\lambda}_{\frac{N-1}{2}}$ which solve exactly the same set of equations as $\lambda_1, \ldots, \lambda_{\frac{N-1}{2}}$.

*Remark* 5. It is natural to expect that the ground states in the sector $\frac{N}{2}$ (for $N$ even) or $\frac{N-1}{2}$ (for $N$ odd) have no hole in their distribution of Bethe roots. However, this hypothesis is not essential for our purpose (computation of the correlation functions in the thermodynamic limit): we essentially build our study on the replacement of sums by integrals as in (55), and the holes contribute only to sub-leading orders to (55). In fact, it is neither essential for our purpose to know the precise sector $R$ of the ground state, since the replacement (55) remains valid for all states given by $R$ real roots with $R$ of order $N/2$ in the thermodynamic limit. Hence we do not have to distinguish further between even and odd $N$.

As in the periodic case [17], it is also convenient to consider the inhomogeneous deformation of the ground state when we introduce inhomogeneity parameters $\xi_1, \ldots, \xi_N$ in the model as in (3). For the previous analysis to remain valid, we may for instance restrict ourselves to the consideration of inhomogeneity parameters $\xi_1, \ldots, \xi_N$ such that $\mathfrak{I}(\xi_n) = \eta/2 = -i/2$, $1 \leq n \leq N$. In that case, we have to define

$$p_{\text{tot}}(\lambda) = \frac{1}{N} \sum_{n=1}^{N} p(\lambda - \xi_n + \eta/2), \tag{67}$$

and it leads to the inhomogeneous density

$$\rho_{\text{tot}}(\lambda) = \frac{1}{N} \sum_{n=1}^{N} \rho(\lambda - \xi_n + \eta/2), \tag{68}$$

solution of the integral equation

$$2\pi \rho_{\text{tot}}(\lambda) - \int_{-\infty}^{\infty} \theta'(\lambda - \mu) \rho_{\text{tot}}(\mu) \, d\mu = p'_{\text{tot}}(\lambda). \tag{69}$$

## 5 Finite-size correlation functions

In this section we explain how to compute the correlation functions, or more precisely the elementary buildings blocks of these correlation functions[13] in the model in finite volume

---

[12]This hypothesis is supported by the fact that the Bethe equations (48) coincide with the Bethe equations of the $\sigma^z$-twisted case [113], a case that can be obtained by a continuous variation of the twist from the periodic case.

[13]These are also called the matrix elements of the density matrix of a chain segment of length $m$.

starting from the SoV solution presented in Section 3. In particular, given $|Q_\tau\rangle$ an eigenstate of the anti-periodic transfer matrix, we consider matrix elements of the form

$$F_{n,n+m-1}(\tau,\epsilon) = \frac{\langle Q_\tau | \prod_{j=1}^m E_{n+j-1}^{\epsilon_{2j-1},\epsilon_{2j}} |Q_\tau\rangle}{\langle Q_\tau | Q_\tau\rangle}, \tag{70}$$

for any $\epsilon \equiv (\epsilon_1, \epsilon_2, \ldots, \epsilon_{2m}) \in \{1,2\}^{2m}$. Here $E^{\epsilon_1,\epsilon_2}$, $\epsilon_1, \epsilon_2 \in \{1,2\}$, stands for the $2 \times 2$ elementary matrix with matrix elements $(E^{\epsilon_1,\epsilon_2})_{i,j} = \delta_{i,\epsilon_1}\delta_{j,\epsilon_2}$. We explain how to compute the matrix elements (70) in a convenient form for the consideration of the homogeneous limit, and also for the consideration of the thermodynamic limit which will be taken in the next section.

As in the periodic case [19], we use the solution of the quantum inverse problem [17,18] to reconstruct the elementary matrices acting on the $n$-th site of the chain as some elements of the monodromy matrix dressed by a product of anti-periodic transfer matrices evaluated at the inhomogeneity parameters. It is indeed easy to show that [103,111]:

**Proposition 5.1.** *Let $E_n^{\epsilon_1,\epsilon_2} \in \mathrm{End}\, V_n$, $(\epsilon_1,\epsilon_2) \in \{1,2\}^2$, be an elementary matrix acting on the $n$-th site of the chain. Then*

$$E_n^{\epsilon_1,\epsilon_2} = \prod_{k=1}^{n-1} \mathcal{T}(\xi_k) \cdot \left[\sigma^x\, T(\xi_n)\right]_{\epsilon_2,\epsilon_1} \cdot \prod_{k=1}^n [\mathcal{T}(\xi_k)]^{-1}$$

$$= \prod_{k=1}^{n-1} \mathcal{T}(\xi_k) \cdot [T(\xi_n)]_{3-\epsilon_2,\epsilon_1} \cdot \prod_{k=1}^n [\mathcal{T}(\xi_k)]^{-1}. \tag{71}$$

Hence, the mean value on an eigenstate (27) of a product of such elementary operators at adjacent sites is given by

$$\langle Q_\tau | \prod_{j=1}^m E_{n+j-1}^{\epsilon_{2j-1},\epsilon_{2j}} |Q_\tau\rangle = \frac{\prod_{k=1}^{n-1} \tau(\xi_k)}{\prod_{k=1}^{n+m-1} \tau(\xi_k)}$$

$$\times \langle Q_\tau | T_{3-\epsilon_{2n},\epsilon_{2n-1}}(\xi_n) \ldots T_{3-\epsilon_{2(n+m-1)},\epsilon_{2(n+m-1)-1}}(\xi_{n+m}) |Q_\tau\rangle, \tag{72}$$

so that, to have access to the correlation functions, it is enough to compute the generic action of a product of elements of the monodromy matrix on an eigenstate and take the resulting scalar product.

Note that, as in the periodic case [19], the only effect of a translation on the chain is a numerical factor given by a product of the corresponding transfer matrix eigenvalues so that, for simplicity, we shall for now on restrict our study to matrix elements of the form

$$F_m(\tau,\epsilon) \equiv F_{1,m}(\tau,\epsilon) = \frac{\langle Q_\tau | \prod_{j=1}^m E_j^{\epsilon_{2j-1},\epsilon_{2j}} |Q_\tau\rangle}{\langle Q_\tau | Q_\tau\rangle} \tag{73}$$

$$= \frac{\langle Q_\tau | T_{3-\epsilon_2,\epsilon_1}(\xi_1) \ldots T_{3-\epsilon_{2m},\epsilon_{2m-1}}(\xi_m) |Q_\tau\rangle}{\prod_{k=1}^m \tau(\xi_k) \langle Q_\tau | Q_\tau\rangle}. \tag{74}$$

Let us also remark that, due to the fact that each eigenstate $|Q_\tau\rangle$ of the anti-periodic transfer matrix is also an eigenstate of the operator $\Gamma^x = \otimes_{n=1}^N \sigma_n^x$ (see (47)), one has the following relation between elementary blocks:

$$F_m(\tau,\epsilon) = \frac{\langle Q_\tau | \Gamma^x \prod_{j=1}^m E_j^{\epsilon_{2j-1},\epsilon_{2j}} \Gamma^x |Q_\tau\rangle}{\langle Q_\tau | Q_\tau\rangle} = \frac{\langle Q_\tau | \prod_{j=1}^m E_j^{3-\epsilon_{2j-1},3-\epsilon_{2j}} |Q_\tau\rangle}{\langle Q_\tau | Q_\tau\rangle}$$

$$= F_m(\tau, 3-\epsilon), \tag{75}$$

in which the $2m$-tuple $3 - \epsilon$ is defined in terms of the $2m$-tuple $\epsilon \equiv (\epsilon_1, \ldots, \epsilon_{2m})$ as $3 - \epsilon \equiv (3 - \epsilon_1, \ldots, 3 - \epsilon_{2m})$.

## 5.1 Left action on separate states

In this section we compute the generic action of a product of matrix elements of the monodromy matrix on a left separate state $\langle Q |$ of the form

$$\langle Q | = \sum_{\mathbf{h} \in \{0,1\}^N} \prod_{n=1}^N Q(\xi_n^{(h_n)}) \, V(\xi_1^{(h_1)}, \dots, \xi_N^{(h_N)}) \langle \mathbf{h} |, \tag{76}$$

where $Q(\lambda) = \prod_{k=1}^R (\lambda - q_k)$ is a polynomial of degree $R \le N$ (not necessarily a solution of the TQ-equation (25)). Our starting point is the action of the monodromy matrix elements $D(\lambda), C(\lambda), B(\lambda)$ (15)-(17) and $A(\lambda)$ (22) on the left SoV basis.

*Remark* 6. Instead of computing the action on a state of the form (76) using (15)-(17) and (22), we could alternatively try to compute the multiple action of a product of transfer matrix elements directly on a Bethe-type state of the form (31) using the Yang-Baxter commutation relations, in the spirit of what is done for model solvable by Bethe Ansatz [19]. However, the fact that the transfer matrix eigenstates can be re-expressed as Bethe-type states involving the multiple action of an *element of the monodromy matrix* as in (31)-(32) is not completely general in the SoV approach, but rather a specificity of models for which the Q-functions have the same functional form as the transfer matrix eigenfunctions of the model: for instance, it is not true in the anti-periodic XXZ model, for which the Q-functions have a double periodicity with respect to the transfer matrix eigenfunctions of the model [110, 131]. So as to remain as general as possible, it is therefore better to start directly from (76) and (15)-(17), (22).

For our purpose, since we need ultimately to evaluate this action only at the inhomogeneity parameters (see (71)), it is in fact more convenient to consider instead of $T_{\epsilon, \epsilon'}(\lambda)$ the operators $\bar{T}_{\epsilon, \epsilon'}(\lambda)$ defined as

$$\bar{T}_{\epsilon, \epsilon'}(\lambda) = \begin{cases} D^{-1}(\lambda + \eta) \, C(\lambda + \eta) \, B(\lambda) & \text{if } (\epsilon, \epsilon') = (1, 1), \\ T_{\epsilon, \epsilon'}(\lambda) & \text{otherwise.} \end{cases} \tag{77}$$

Indeed, since $\det_q T(\xi_i + \eta) = 0$, it follows from (9) that

$$\bar{T}_{\epsilon, \epsilon'}(\xi_i) = T_{\epsilon, \epsilon'}(\xi_i) \qquad \forall i \in \{1, \dots, N\}, \quad \forall \epsilon, \epsilon' \in \{1, 2\}, \tag{78}$$

so that the formula (71) can be written in terms of the matrix elements $\bar{T}_{\epsilon, \epsilon'}$ instead of $T_{\epsilon, \epsilon'}$. Note that (77) is well defined as soon as $\lambda \notin \{\xi_i - \eta, \xi_i - 2\eta \mid i = 1, \dots, N\}$ since $D(\lambda)$ is invertible for any $\lambda \ne \xi_i, \xi_i - \eta, i = 1, \dots N$. The action of $\bar{A}(\lambda) \equiv \bar{T}_{1,1}(\lambda)$ on a SoV state $\langle \mathbf{h} |$ is then slightly simpler than the action of $A(\lambda)$ (22).

It is easy to compute the action of the operators $\bar{T}_{\epsilon, \epsilon'}(\lambda)$ on the separate state (76). We obtain

$$\langle Q | D(\lambda) = \sum_{\mathbf{h}} d_{\mathbf{h}}(\lambda) \prod_{n=1}^N Q(\xi_n^{(h_n)}) V(\xi_1^{(h_1)}, \dots, \xi_N^{(h_N)}) \langle \mathbf{h} |, \tag{79}$$

$$\begin{aligned} \langle Q | B(\lambda) = & -\sum_{b=1}^N a(\xi_b) \sum_{\mathbf{h}} \delta_{h_b, 1} \prod_{n=1}^N Q(\xi_n^{(h_n)}) \prod_{\substack{n=1 \\ n \ne b}}^N \frac{\lambda - \xi_n^{(h_n)}}{\xi_b^{(1)} - \xi_n^{(h_n)}} V(\xi_1^{(h_1)}, \dots, \xi_N^{(h_N)}) \langle \mathrm{T}_b^- \mathbf{h} | \\ = & -\sum_{b=1}^N \frac{a(\xi_b)}{\lambda - \xi_b} \frac{Q(\xi_b - \eta)}{Q(\xi_b)} \sum_{\mathbf{h}} \delta_{h_b, 0} \frac{d_{\mathbf{h}}(\lambda) \prod_{n=1}^N Q(\xi_n^{(h_n)})}{\prod_{n \ne b} (\xi_b - \xi_n^{(h_n)})} \\ & \times V(\xi_1^{(h_1)}, \dots, \xi_N^{(h_N)}) \langle \mathbf{h} |, \tag{80} \end{aligned}$$

$$\langle Q | C(\lambda) = \sum_{b=1}^{N} d(\xi_b^{(1)}) \sum_{\mathbf{h}} \delta_{h_b,0} \prod_{n=1}^{N} Q(\xi_n^{(h_n)}) \prod_{\substack{n=1 \\ n \neq b}}^{N} \frac{\lambda - \xi_n^{(h_n)}}{\xi_b - \xi_n^{(h_n)}} V(\xi_1^{(h_1)}, \dots, \xi_N^{(h_N)}) \langle T_b^+ \mathbf{h} |$$

$$= \sum_{b=1}^{N} \frac{d(\xi_b^{(1)})}{\lambda - \xi_b^{(1)}} \frac{Q(\xi_b)}{Q(\xi_b - \eta)} \sum_{\mathbf{h}} \delta_{h_b,1} \frac{d_{\mathbf{h}}(\lambda) \prod_{n=1}^{N} Q(\xi_n^{(h_n)})}{\prod_{n \neq b}(\xi_b - \xi_n^{(h_n)})}$$
$$\times V(\xi_1^{(h_1)}, \dots, \xi_N^{(h_N)}) \langle \mathbf{h} |, \qquad (81)$$

and a similar (although more involved) expression can be obtained for the action of $\bar{A}(\lambda)$ on $\langle Q |$.

It is obviously possible, from these formulas, to compute the multiple action of any string of operators $\bar{T}_{\epsilon_2,\epsilon_1}(\lambda_1) \bar{T}_{\epsilon_4,\epsilon_3}(\lambda_2) \dots \bar{T}_{\epsilon_{2m},\epsilon_{2m-1}}(\lambda_m)$ on the state $\langle Q |$ as a multiple sum over choices of inhomogeneity parameters along the chain, but such an expression would not be convenient for the consideration of the homogeneous limit. We therefore now explain how to write this action in terms of a multiple contour integral that we can transform into a more convenient form for the consideration of the homogeneous limit. In fact, one can show the following result:

**Proposition 5.1.** *Let $\lambda$ be a generic parameter. The left action of the operator $\bar{T}_{\epsilon_2,\epsilon_1}(\lambda)$, $\epsilon_1, \epsilon_2 \in \{1,2\}$, on a generic separate state $\langle Q |$ of the form (76) can be written as the following sum of contour integrals:*

$$\langle Q | \bar{T}_{\epsilon_2,\epsilon_1}(\lambda) = \sum_{\mathbf{h}} d_{\mathbf{h}}(\lambda) \prod_{n=1}^{N} Q(\xi_n^{(h_n)}) \left( -\oint_{\Gamma_2} \frac{dz_2}{2\pi i (\lambda - z_2)} \frac{a(z_2)}{d_{\mathbf{h}}(z_2)} \frac{Q(z_2 - \eta)}{Q(z_2)} \right)^{2 - \epsilon_2}$$
$$\times \left( \oint_{\Gamma_1} \frac{dz_1}{2\pi i (\lambda - z_1)} \frac{d(z_1)}{d_{\mathbf{h}}(z_1)} \frac{Q(z_1 + \eta)}{Q(z_1)} \right)^{2 - \epsilon_1} \left( \frac{z_1 - z_2}{z_1 - z_2 + \eta} \right)^{(2 - \epsilon_1)(2 - \epsilon_2)}$$
$$\times V(\xi_1^{(h_1)}, \dots, \xi_N^{(h_N)}) \langle \mathbf{h} |, \quad (82)$$

*in which the contour $\Gamma_2$ surrounds counterclockwise the points $\xi_n$, $1 \leq n \leq N$, and no other poles in the integrand, whereas the contour $\Gamma_1$ surrounds counterclockwise the points $\xi_n - \eta$, $1 \leq n \leq N$, the point $z_2 - \eta$ if $\epsilon_2 = 1$, and no other poles in the integrand.*

*Similarly, for generic parameters $\lambda_1, \dots, \lambda_m$, the multiple action of a product of operators $\bar{T}_{\epsilon_2,\epsilon_1}(\lambda_1) \bar{T}_{\epsilon_4,\epsilon_3}(\lambda_2) \dots \bar{T}_{\epsilon_{2m},\epsilon_{2m-1}}(\lambda_m)$, $\epsilon_i \in \{1,2\}$, $1 \leq i \leq 2m$, on a generic separate state $\langle Q |$ of the form (76) can be written as the following sum of contour integrals:*

$$\langle Q | \bar{T}_{\epsilon_2,\epsilon_1}(\lambda_1) \bar{T}_{\epsilon_4,\epsilon_3}(\lambda_2) \dots \bar{T}_{\epsilon_{2m},\epsilon_{2m-1}}(\lambda_m) = \sum_{\mathbf{h}} \prod_{j=1}^{m} d_{\mathbf{h}}(\lambda_j) \prod_{n=1}^{N} Q(\xi_n^{(h_n)})$$

$$\times \prod_{j=m}^{1} \left[ \left( -\oint_{\Gamma_{2j}} \frac{dz_{2j}}{2\pi i (\lambda_j - z_{2j})} \frac{a(z_{2j})}{d_{\mathbf{h}}(z_{2j})} \frac{Q(z_{2j} - \eta)}{Q(z_{2j})} \prod_{k=1}^{j-1} \frac{z_{2j} - \lambda_k - \eta}{z_{2j} - \lambda_k} \right)^{2 - \epsilon_{2j}} \right.$$

$$\left. \times \left( \oint_{\Gamma_{2j-1}} \frac{dz_{2j-1}}{2\pi i (\lambda_j - z_{2j-1})} \frac{d(z_{2j-1})}{d_{\mathbf{h}}(z_{2j-1})} \frac{Q(z_{2j-1} + \eta)}{Q(z_{2j-1})} \prod_{k=1}^{j-1} \frac{z_{2j-1} - \lambda_k + \eta}{z_{2j-1} - \lambda_k} \right)^{2 - \epsilon_{2j-1}} \right]$$

$$\times \prod_{1 \leq j < k \leq 2m} \left( \frac{z_j - z_k}{z_j - z_k + (-1)^k \eta} \right)^{(2 - \epsilon_j)(2 - \epsilon_k)} V(\xi_1^{(h_1)}, \dots, \xi_N^{(h_N)}) \langle \mathbf{h} |, \quad (83)$$

*in which the contours $\Gamma_{2j}$ surround counterclockwise the points $\xi_n$, $1 \leq n \leq N$, the points $z_{2k-1} + \eta$, $k > j$, and no other poles in the integrand, whereas the contours $\Gamma_{2j-1}$ surround counterclockwise the points $\xi_n - \eta$, $1 \leq n \leq N$, the points $z_{2k} - \eta$, $k \geq j$, and no other poles in the integrand.*

*Proof.* The expression (82) clearly coincides with (79) in the case $(\epsilon_2, \epsilon_1) = (2,2)$.

Let us now consider the action (80) of $\bar{T}_{1,2}(\lambda) = B(\lambda)$ on $\langle Q|$. The idea is to see the sum as the development of an integral around a contour by the residue theorem, which leads to the identity

$$\langle Q|B(\lambda) = -\oint_{\Gamma(\{\xi_n\}_{n=1\to N})} \frac{dz}{2\pi i} \frac{a(z)}{\lambda - z} \frac{Q(z-\eta)}{Q(z)}$$
$$\times \sum_{\mathbf{h}} \frac{d_{\mathbf{h}}(\lambda)}{d_{\mathbf{h}}(z)} \prod_{n=1}^{N} Q(\xi_n^{(h_n)}) \, V(\xi_1^{(h_1)}, \ldots, \xi_N^{(h_N)}) \langle \mathbf{h}|, \quad (84)$$

where the contour $\Gamma(\{\xi_n\}_{n=1\to N})$ surrounds counterclockwise the points $\xi_n$, $1 \le n \le N$, and no other pole of the integrand. This result coincides with (82) for $(\epsilon_2, \epsilon_1) = (1,2)$.

We can proceed similarly for the action of $\bar{T}_{2,1}(\lambda) = C(\lambda)$, rewriting (81) as an integral around a contour by the residue theorem, which leads to the identity

$$\langle Q|C(\lambda) = \oint_{\Gamma(\{\xi_n-\eta\}_{n=1\to N})} \frac{dz}{2\pi i} \frac{d(z)}{\lambda - z} \frac{Q(z+\eta)}{Q(z)}$$
$$\times \sum_{\mathbf{h}} \frac{d_{\mathbf{h}}(\lambda)}{d_{\mathbf{h}}(z)} \prod_{n=1}^{N} Q(\xi_n^{(h_n)}) \, V(\xi_1^{(h_1)}, \ldots, \xi_N^{(h_N)}) \langle \mathbf{h}|, \quad (85)$$

with $\Gamma(\{\xi_n - \eta\}_{n=1\to N})$ surrounding counterclockwise the points $\xi_n - \eta$, $1 \le n \le N$, and no other pole of the integrand. This result coincides with (82) for $(\epsilon_2, \epsilon_1) = (2,1)$.

Finally, let us consider the action of $\bar{T}_{1,1}(\lambda) = \bar{A}(\lambda)$ on $\langle Q|$, which is the more involved one, as it requires to compute the successive action of $D^{-1}(\lambda + \eta)$, $C(\lambda + \eta)$ and $B(\lambda)$ on the state $\langle Q|$. Using (15) and (16), one can write

$$\langle Q|\bar{A}(\lambda) = \sum_{b=1}^{N} \frac{d(\xi_b - \eta)}{\lambda - \xi_b + \eta} \frac{Q(\xi_b)}{Q(\xi_b - \eta)} \sum_{\mathbf{h}} \delta_{h_b,1} \prod_{n=1}^{N} Q(\xi_n^{(h_n)})$$
$$\times \frac{V(\xi_1^{(h_1)}, \ldots, \xi_n^{(h_n)})}{\prod_{\ell \ne b}(\xi_b^{(1)} - \xi_\ell^{(h_\ell)})} \langle \mathbf{h}|B(\lambda), \quad (86)$$

which corresponds to the evaluation by the sum over the residues of the following contour integral:

$$\langle Q|\bar{A}(\lambda) = \oint_{\Gamma(\{\xi_n-\eta\}_{n=1\to N})} \frac{dz}{2\pi i} \frac{d(z)}{\lambda - z} \frac{Q(z+\eta)}{Q(z)}$$
$$\times \sum_{\mathbf{h}} \frac{V(\xi_1^{(h_1)}, \ldots, \xi_n^{(h_N)})}{d_{\mathbf{h}}(z)} \prod_{n=1}^{N} Q(\xi_n^{(h_n)}) \langle \mathbf{h}|B(\lambda). \quad (87)$$

Using now (17), we obtain

$$
\langle Q | \bar{A}(\lambda) = -\sum_{b=1}^{N} a(\xi_b) \oint_{\Gamma(\{\xi_n-\eta\}_{n=1\to N})} \frac{dz}{2\pi i} \frac{d(z)}{\lambda-z} \frac{Q(z+\eta)}{Q(z)} \frac{z-\xi_b}{z-\xi_b+\eta} \frac{Q(\xi_b-\eta)}{Q(\xi_b)}
$$

$$
\times \sum_{\mathbf{h}} \delta_{h_b,0} \prod_{\ell\neq b} \frac{\lambda-\xi_\ell^{(h_\ell)}}{\xi_b-\xi_\ell^{(h_\ell)}} \frac{V(\xi_1^{(h_1)},\dots,\xi_n^{(h_n)})}{d_{\mathbf{h}}(z)} \prod_{n=1}^{N} Q(\xi_n^{(h_n)}) \langle \mathbf{h} |
$$

$$
= -\oint_{\Gamma(\{\xi_n\}_{n=1\to N})} \frac{dz'}{2\pi i} \frac{a(z')}{\lambda-z'} \frac{Q(z'-\eta)}{Q(z')} \oint_{\Gamma(\{\xi_n-\eta\}_{n=1\to N}\cup\{z'-\eta\})} \frac{dz}{2\pi i} \frac{d(z)}{\lambda-z}
$$

$$
\times \frac{Q(z+\eta)}{Q(z)} \frac{z-z'}{z-z'+\eta} \sum_{\mathbf{h}} \frac{d_{\mathbf{h}}(\lambda)}{d_{\mathbf{h}}(z) \, d_{\mathbf{h}}(z')} \prod_{n=1}^{N} Q(\xi_n^{(h_n)}) V(\xi_1^{(h_1)},\dots,\xi_n^{(h_n)}) \langle \mathbf{h} |, \qquad (88)
$$

in which we have again used the residue theorem to recast the sum as a contour integral over $z'$. Note that doing this the pole at $\xi_b - \eta$ becomes a pole at $z' - \eta$, hence we have to deform the contour of the integral over $z$ to take into account the residue at this pole. The expression (88) coincides with (82) in the case $(\epsilon_2, \epsilon_1) = (1, 1)$.

The general result is then obtained by induction along the same lines. $\qquad\square$

The multiple integral representation (83) of Proposition 5.1 can easily be recast in a more convenient form for the further consideration of the homogeneous limit.

**Proposition 5.2.** *For generic parameters $\lambda_1, \dots, \lambda_m$, the multiple action of a product of operators $\bar{T}_{\epsilon_2,\epsilon_1}(\lambda_1) \bar{T}_{\epsilon_4,\epsilon_3}(\lambda_2) \dots \bar{T}_{\epsilon_{2m},\epsilon_{2m-1}}(\lambda_m)$, $\epsilon_i \in \{1,2\}$, $1 \le i \le 2m$, on a generic separate state $\langle Q |$ of the form (76) can be written as the following sum of contour integrals:*

$$
\langle Q | \bar{T}_{\epsilon_2,\epsilon_1}(\lambda_1) \bar{T}_{\epsilon_4,\epsilon_3}(\lambda_2) \dots \bar{T}_{\epsilon_{2m},\epsilon_{2m-1}}(\lambda_m) = \sum_{\mathbf{h}} \prod_{j=1}^{m} d_{\mathbf{h}}(\lambda_j) \prod_{n=1}^{N} Q(\xi_n^{(h_n)})
$$

$$
\times \prod_{j=m}^{1} \left[ \left( -\oint_{\mathcal{C}_j^\infty} \frac{dz_{2j}}{2\pi i (z_{2j}-\lambda_j)} \frac{a(z_{2j})}{d_{\mathbf{h}}(z_{2j})} \frac{Q(z_{2j}-\eta)}{Q(z_{2j})} \prod_{k=1}^{j-1} \frac{z_{2j}-\lambda_k-\eta}{z_{2j}-\lambda_k} \right)^{2-\epsilon_{2j}} \right.
$$

$$
\times \left. \left( \oint_{\mathcal{C}_j^\infty} \frac{dz_{2j-1}}{2\pi i (z_{2j-1}-\lambda_j)} \frac{d(z_{2j-1})}{d_{\mathbf{h}}(z_{2j-1})} \frac{Q(z_{2j-1}+\eta)}{Q(z_{2j-1})} \prod_{k=1}^{j-1} \frac{z_{2j-1}-\lambda_k+\eta}{z_{2j-1}-\lambda_k} \right)^{2-\epsilon_{2j-1}} \right]
$$

$$
\times \prod_{1\le j<k\le 2m} \left( \frac{z_j-z_k}{z_j-z_k+(-1)^k\eta} \right)^{(2-\epsilon_j)(2-\epsilon_k)} V(\xi_1^{(h_1)},\dots,\xi_N^{(h_N)}) \langle \mathbf{h} |, \quad (89)
$$

*where the contours $\mathcal{C}_j^\infty$ $1 \le j \le 2m$, surround counterclockwise the points $q_n$, $1 \le n \le R$, $\lambda_\ell$, $1 \le \ell \le j$, the pole at infinity, and no other pole of the integrand.*

*Proof.* Let us prove by recursion on $n$ the formula

$$
\langle Q | \bar{T}_{\epsilon_2,\epsilon_1}(\lambda_1) \bar{T}_{\epsilon_4,\epsilon_3}(\lambda_2) \dots \bar{T}_{\epsilon_{2m},\epsilon_{2m-1}}(\lambda_m) = \sum_{\mathbf{h}} \prod_{j=1}^{m} d_{\mathbf{h}}(\lambda_j) \prod_{n=1}^{N} Q(\xi_n^{(h_n)})
$$

$$
\times \prod_{j=m}^{n} \left[ \left( -\oint_{\Gamma_{2j}} \frac{dz_{2j}}{2\pi i \,(\lambda_j - z_{2j})} \frac{a(z_{2j})}{d_{\mathbf{h}}(z_{2j})} \frac{Q(z_{2j} - \eta)}{Q(z_{2j})} \prod_{k=1}^{j-1} \frac{z_{2j} - \lambda_k - \eta}{z_{2j} - \lambda_k} \right)^{2-\epsilon_{2j}} \right.
$$

$$
\times \left. \left( \oint_{\Gamma_{2j-1}} \frac{dz_{2j-1}}{2\pi i \,(\lambda_j - z_{2j-1})} \frac{d(z_{2j-1})}{d_{\mathbf{h}}(z_{2j-1})} \frac{Q(z_{2j-1} + \eta)}{Q(z_{2j-1})} \prod_{k=1}^{j-1} \frac{z_{2j-1} - \lambda_k + \eta}{z_{2j-1} - \lambda_k} \right)^{2-\epsilon_{2j-1}} \right]
$$

$$
\times \prod_{j=n-1}^{1} \left[ \left( \oint_{\mathcal{C}_j^{\infty}} \frac{dz_{2j}}{2\pi i \,(\lambda_j - z_{2j})} \frac{a(z_{2j})}{d_{\mathbf{h}}(z_{2j})} \frac{Q(z_{2j} - \eta)}{Q(z_{2j})} \prod_{k=1}^{j-1} \frac{z_{2j} - \lambda_k - \eta}{z_{2j} - \lambda_k} \right)^{2-\epsilon_{2j}} \right.
$$

$$
\times \left. \left( -\oint_{\mathcal{C}_j^{\infty}} \frac{dz_{2j-1}}{2\pi i \,(\lambda_j - z_{2j-1})} \frac{d(z_{2j-1})}{d_{\mathbf{h}}(z_{2j-1})} \frac{Q(z_{2j-1} + \eta)}{Q(z_{2j-1})} \prod_{k=1}^{j-1} \frac{z_{2j-1} - \lambda_k + \eta}{z_{2j-1} - \lambda_k} \right)^{2-\epsilon_{2j-1}} \right]
$$

$$
\times \prod_{1 \leq j < k \leq 2m} \left( \frac{z_j - z_k}{z_j - z_k + (-1)^k \eta} \right)^{(2-\epsilon_j)(2-\epsilon_k)} V(\xi_1^{(h_1)}, \dots, \xi_N^{(h_N)}) \langle \mathbf{h} |, \quad (90)
$$

which coincides with (83) for $n = 1$ and with (89) for $n = m$.

Let us suppose that (90) holds for a given $n$, $1 \leq n < m$, and let us rewrite the integral over $z_{2n-1}$ using the poles outside of the integration contour $\Gamma_{2n-1}$. These poles are at the zeroes $q_1, \dots, q_R$ of $Q$, at $\lambda_j$ for $j < n$ and at infinity. Note that the apparent poles at $\xi_j$, $1 \leq j \leq N$, are in fact regular points due to the factor $d(z_{2n-1})$ in the numerator. Similarly, the poles at $z_{2k-1} + \eta$ for $k > n$ are also regular points since the integral over $z_{2k-1}$ has to be finally evaluated by its residue at $z_{2k-1} = \xi_\ell - \eta$ for some $\ell \in \{1, \dots, N\}$. Finally, the apparent poles at $z_j - \eta$ for $j < 2n - 1$ are also regular points since the integral over $z_j$ is first evaluated by its residues at $\infty$ (and the corresponding factor disappears), at a roots $q_k$ of $Q$ (and the factor $Q(z_{2n-1} + \eta)$ in the numerator vanishes) or at $\lambda_j$ for $j < n$ (and the factor $z_{2j-1} - \lambda_j + \eta$ in the numerator vanishes). Hence the integral over $z_{2n-1}$ can be rewritten as a contour integral surrounding the points $q_1, \dots, q_R$, $\lambda_j$ for $j < n$, and $\infty$ with index $-1$. One then consider the integral over $z_{2n}$ and show similarly that the points $\xi_j - \eta$, $1 \leq j \leq N$, $z_{2k} - \eta$, $k > n$, and $z_\ell + \eta$, $\ell < 2n$, are regular points, so that the integral can be written as a contour integral around the poles at $q_1, \dots, q_R$, $\lambda_j$ for $j < n$, and $\infty$ with index $-1$. Hence the representation (90) holds also for $n + 1$. □

The integral representation (89) can be evaluated as a sum over its residues, which leads to

**Corollary 5.1.** *The multiple action of a product of operators $\bar{T}_{\epsilon_2,\epsilon_1}(\lambda_1) \bar{T}_{\epsilon_4,\epsilon_3}(\lambda_2) \dots \bar{T}_{\epsilon_{2m},\epsilon_{2m-1}}(\lambda_m)$, $\epsilon_i \in \{1, 2\}$, $1 \leq i \leq 2m$, on a generic separate state $\langle Q |$ of the form (76) can be written as a sum over separate states of the form (76) as*

$$
\langle Q | \bar{T}_{\epsilon_2,\epsilon_1}(\lambda_1) \bar{T}_{\epsilon_4,\epsilon_3}(\lambda_2) \dots \bar{T}_{\epsilon_{2m},\epsilon_{2m-1}}(\lambda_m) = \sum_{n_\infty=0}^{m_\epsilon} (-1)^{(m - m_\epsilon + n_\infty)N}
$$

$$
\times \left[ \langle Q | \bar{T}_{\epsilon_2,\epsilon_1}(\lambda_1) \bar{T}_{\epsilon_4,\epsilon_3}(\lambda_2) \dots \bar{T}_{\epsilon_{2m},\epsilon_{2m-1}}(\lambda_m) \right]_{n_\infty}, \quad (91)
$$

*where*

$$
\left[ \langle Q | \bar{T}_{\epsilon_2,\epsilon_1}(\lambda_1) \, \bar{T}_{\epsilon_4,\epsilon_3}(\lambda_2) \dots \bar{T}_{\epsilon_{2m},\epsilon_{2m-1}}(\lambda_m) \right]_{n_\infty}
$$

$$
= \sum_{(\bar{\epsilon}_1,\dots,\bar{\epsilon}_{2m}) \in \mathcal{E}_{\epsilon,n_\infty}} \sum_{a_1=1}^{(R+1)\bar{\epsilon}_1} \sum_{\substack{a_2=1 \\ a_2 \neq a_1}}^{(R+1)\bar{\epsilon}_2} \cdots \sum_{\substack{a_{2m-1}=1 \\ a_{2m-1} \notin \{a_1,\dots,a_{2m-2}\}}}^{(R+m)\bar{\epsilon}_{2m-1}} \sum_{\substack{a_{2m}=1 \\ a_{2m} \notin \{a_1,\dots,a_{2m-1}\}}}^{(R+m)\bar{\epsilon}_{2m}}
$$

$$
\times \prod_{j=1}^{m} \left( \frac{d(q_{a_{2j-1}}) \prod_{k=1}^{R+j-1}(q_{a_{2j-1}} - q_k + \eta)}{\prod_{\substack{k=1 \\ k \neq a_{2j-1}}}^{R+j} (q_{a_2j-1} - q_k)} \right)^{\bar{\epsilon}_{2j-1}} \left( -\frac{a(q_{a_{2j}}) \prod_{k=1}^{R+j-1}(q_{a_{2j}} - q_k - \eta)}{\prod_{\substack{k=1 \\ k \neq a_{2j}}}^{R+j} (q_{a_2j} - q_k)} \right)^{\bar{\epsilon}_{2j}}
$$

$$
\times \prod_{1 \leq j < k \leq 2m} \left( \frac{q_{a_j} - q_{a_k}}{q_{a_j} - q_{a_k} + (-1)^k \eta} \right)^{\bar{\epsilon}_j \bar{\epsilon}_k} \langle \bar{Q}_{a,\bar{\epsilon}}^\lambda | . \quad (92)
$$

*Here we have defined, for a given 2m-tuple $\epsilon \equiv (\epsilon_1,\dots,\epsilon_{2m})$,*

$$
m_\epsilon = \sum_{j=1}^{2m} (2 - \epsilon_j), \quad (93)
$$

$$
\mathcal{E}_{\epsilon,n_\infty} = \left\{ (\bar{\epsilon}_1,\dots,\bar{\epsilon}_{2m}) \in \{0,1\}^N \mid \bar{\epsilon}_j \leq 2 - \epsilon_j \text{ and } \sum_{j=1}^{2m} \bar{\epsilon}_j = m_\epsilon - n_\infty \right\}. \quad (94)
$$

*Moreover, we have used the shortcut notation*

$$
q_{R+j} = \lambda_j, \quad 1 \leq j \leq m, \quad (95)
$$

*and $\bar{Q}_{a,\bar{\epsilon}}^\lambda$ is a polynomial of degree $R + m - m_\epsilon + n_\infty$ defined in terms of Q, of the $\lambda_k$, $1 \leq k \leq m$, and of the $a_j$ and the $\bar{\epsilon}_j$ ($1 \leq j \leq 2m$) as*

$$
\bar{Q}_{a,\bar{\epsilon}}^\lambda(\lambda) = Q(\lambda) \frac{\prod_{j=1}^m (\lambda - \lambda_j)}{\prod_{j=1}^{2m} (\lambda - q_{a_j})^{\bar{\epsilon}_j}} = \frac{\prod_{j=1}^{R+m} (\lambda - q_j)}{\prod_{j=1}^{2m} (\lambda - q_{a_j})^{\bar{\epsilon}_j}}. \quad (96)
$$

*Proof.* We are just writing the development of the multiple contour integrals (89) in terms of the sum on the residues. Here $0 \leq n_\infty \leq m_\epsilon$ corresponds to the number of residues at infinity that we take so that we are organizing these sums w.r.t. $n_\infty$. □

Note that, in the expression (91)-(92), we can now particularize the parameters $\lambda_i$, $1 \leq i \leq m$, to be equal to some inhomogeneity parameters. We can therefore directly use (91)-(92) to express the matrix elements of the form (73).

## 5.2 Multiple sum representation for the correlation functions in finite volume

As a consequence of the results of the previous subsection, we can now write any matrix elements of the form (73) as a sum over scalar products of separate states:

$$
F_m(\tau, \epsilon) = \prod_{k=1}^{m} \frac{1}{\tau(\xi_k)} \sum_{n_\infty=0}^{m'_\epsilon} (-1)^{(m-m_{\epsilon'}+n_\infty)N}
$$

$$
\times \sum_{(\bar{\epsilon}_1,\ldots,\bar{\epsilon}_{2m})\in\mathcal{E}_{\epsilon',n_\infty}} \sum_{a_1=1}^{(R+1)\bar{\epsilon}_1} \sum_{\substack{a_2=1 \\ a_2\neq a_1}}^{(R+1)\bar{\epsilon}_2} \cdots \sum_{\substack{a_{2m-1}=1 \\ a_{2m-1}\notin\{a_1,\ldots,a_{2m-2}\}}}^{(R+m)\bar{\epsilon}_{2m-1}} \sum_{\substack{a_{2m}=1 \\ a_{2m}\notin\{a_1,\ldots,a_{2m-1}\}}}^{(R+m)\bar{\epsilon}_{2m}}
$$

$$
\times \prod_{j=1}^{m} \left( \frac{d(q_{a_{2j-1}}) \prod_{k=1}^{R+j-1}(q_{a_{2j-1}} - q_k + \eta)}{\prod_{\substack{k=1 \\ k\neq a_{2j-1}}}^{R+j} (q_{a_{2j-1}} - q_k)} \right)^{\bar{\epsilon}_{2j-1}} \left( -\frac{a(q_{a_{2j}}) \prod_{k=1}^{R+j-1}(q_{a_{2j}} - q_k - \eta)}{\prod_{\substack{k=1 \\ k\neq a_{2j}}}^{R+j} (q_{a_{2j}} - q_k)} \right)^{\bar{\epsilon}_{2j}}
$$

$$
\times \prod_{1\leq j<k\leq 2m} \left( \frac{q_{a_j} - q_{a_k}}{q_{a_j} - q_{a_k} + (-1)^k\eta} \right)^{\bar{\epsilon}_j\bar{\epsilon}_k} \frac{\langle \bar{Q}_{a,\bar{\epsilon}}^\xi | Q_\tau \rangle}{\langle Q_\tau | Q_\tau \rangle}, \quad (97)
$$

in which we have defined the $2m$-tuple $\epsilon' \equiv (\epsilon'_1,\ldots,\epsilon'_{2m})$ in terms of the $2m$-tuple $\epsilon \equiv (\epsilon_1,\ldots,\epsilon_{2m})$ by

$$
\epsilon'_{2j-1} = \epsilon_{2j-1}, \qquad \epsilon'_{2j} = 3 - \epsilon_{2j}, \qquad 1 \leq j \leq m, \quad (98)
$$

and defined $m_{\epsilon'}$, $\mathcal{E}_{\epsilon',n_\infty}$ as in (93)-(94) but in terms of $\epsilon'$ rather than $\epsilon$. Similarly as in (99)-(100), we have used the shortcut notations:

$$
q_{R+j} = \xi_j, \quad 1 \leq j \leq m, \quad (99)
$$

and $\bar{Q}_{a,\bar{\epsilon}}^\xi$ is a polynomial of degree $R + m - m_{\epsilon'} + n_\infty$ defined in terms of $Q \equiv Q_\tau$, of the $\xi_k$, $1 \leq k \leq m$, and of the $a_j$ and the $\bar{\epsilon}_j$ ($1 \leq j \leq 2m$) as

$$
\bar{Q}_{a,\bar{\epsilon}}^\xi(\lambda) = Q(\lambda) \frac{\prod_{j=1}^{m}(\lambda - \xi_j)}{\prod_{j=1}^{2m}(\lambda - q_{a_j})^{\bar{\epsilon}_j}} = \frac{\prod_{j=1}^{R+m}(\lambda - q_j)}{\prod_{j=1}^{2m}(\lambda - q_{a_j})^{\bar{\epsilon}_j}}. \quad (100)
$$

We also recall that $R$ is the degree of the polynomial $Q_\tau$.

This expression (97) can be rewritten with similar notations as those used in the periodic case [19].

**Proposition 5.2.** *For a given $2m$-tuple $\epsilon \equiv (\epsilon_1,\ldots,\epsilon_{2m}) \in \{1,2\}^{2m}$, let us define the sets $\alpha_\epsilon^-$ and $\alpha_\epsilon^+$ as*

$$
\alpha_\epsilon^- = \{j : 1 \leq j \leq m, \epsilon_{2j-1} = 1\}, \quad \#\alpha_\epsilon^- = s_\epsilon \quad (101)
$$

$$
\alpha_\epsilon^+ = \{j : 1 \leq j \leq m, \epsilon_{2j} = 2\}, \qquad \#\alpha_\epsilon^+ = s'_\epsilon. \quad (102)
$$

*Then,*

$$F_m(\tau, \boldsymbol{\epsilon}) = \prod_{k=1}^{m} \frac{1}{\tau(\xi_k)} \sum_{\substack{\bar{\alpha}_{\boldsymbol{\epsilon}}^{-} \subset \alpha_{\boldsymbol{\epsilon}}^{-} \\ \bar{\alpha}_{\boldsymbol{\epsilon}}^{+} \subset \alpha_{\boldsymbol{\epsilon}}^{+}}} (-1)^{(m - \#\bar{\alpha}^- + \#\bar{\alpha}^+)N} \sum_{\{a_j, a_j'\}}$$

$$\times \prod_{j \in \bar{\alpha}_{\boldsymbol{\epsilon}}^{-}} \left( \frac{d(q_{a_j}) \prod_{\substack{k=1 \\ k \in \mathbf{A}_j}}^{R+j-1}(q_{a_j} - q_k + \eta)}{\prod_{\substack{k=1 \\ k \in \mathbf{A}_j'}}^{R+j}(q_{a_j} - q_k)} \right) \prod_{j \in \bar{\alpha}_{\boldsymbol{\epsilon}}^{+}} \left( -\frac{a(q_{a_j'}) \prod_{\substack{k=1 \\ k \in \mathbf{A}_j'}}^{R+j-1}(q_k - q_{a_j'} + \eta)}{\prod_{\substack{k=1 \\ k \in \mathbf{A}_{j+1}}}^{R+j}(q_k - q_{a_j'})} \right)$$

$$\times \frac{\langle \bar{Q}_{\mathbf{A}_{m+1}} | Q_\tau \rangle}{\langle Q_\tau | Q_\tau \rangle}. \quad (103)$$

*In* (103), *the first summation is taken over all subsets $\bar{\alpha}_{\boldsymbol{\epsilon}}^-$ of $\alpha_{\boldsymbol{\epsilon}}^-$ and $\bar{\alpha}_{\boldsymbol{\epsilon}}^+$ of $\alpha_{\boldsymbol{\epsilon}}^+$, whereas the second summation is taken over the indices $a_j$ for $j \in \bar{\alpha}_{\boldsymbol{\epsilon}}^-$ and $a_j'$ for $j \in \bar{\alpha}_{\boldsymbol{\epsilon}}^+$ such that*

$$1 \le a_j \le R + j, \quad a_j \in \mathbf{A}_j, \qquad 1 \le a_j' \le R + j, \quad a_j' \in \mathbf{A}_j', \quad (104)$$

*where*

$$\mathbf{A}_j = \{b : 1 \le b \le R + m, b \ne a_k, a_k', k < j\}, \quad (105)$$

$$\mathbf{A}_j' = \{b : 1 \le b \le R + m, b \ne a_k', k < j \text{ and } b \ne a_k, k \le j\}\}. \quad (106)$$

*Moreover, $\bar{Q}_{\mathbf{A}_{m+1}}$ is the polynomial of degree $\#\mathbf{A}_{m+1} = R + m - \#\bar{\alpha}_- - \#\bar{\alpha}_+$ defined in terms of the roots $q_1, \ldots, q_R$ of $Q_\tau$ and of $q_{R+j} \equiv \xi_j$, $1 \le j \le m$, as*

$$\bar{Q}_{\mathbf{A}_{m+1}}(\lambda) = \prod_{j \in \mathbf{A}_{m+1}} (\lambda - q_j). \quad (107)$$

*Remark* 7. The set (102) and (101) are in fact complementary to the set $\alpha^+$ and $\alpha^-$ defined in [19] in the periodic case. One recovers the same sets by considering the sets for $F_m(\tau, 3 - \boldsymbol{\epsilon})$ using the fact that $F_m(\tau, \boldsymbol{\epsilon}) = F_m(\tau, 3 - \boldsymbol{\epsilon})$ (75) due to the $\Gamma^x$ symmetry.

*Remark* 8. The sum over the subsets $\bar{\alpha}_{\boldsymbol{\epsilon}}^-$ and $\bar{\alpha}_{\boldsymbol{\epsilon}}^+$ of $\alpha_{\boldsymbol{\epsilon}}^-$ and $\alpha_{\boldsymbol{\epsilon}}^+$ can be organized as in (97) in terms of the number $n_\infty$ of residues taken at infinity by writing

$$\sum_{\substack{\bar{\alpha}_{\boldsymbol{\epsilon}}^{-} \subset \alpha_{\boldsymbol{\epsilon}}^{-} \\ \bar{\alpha}_{\boldsymbol{\epsilon}}^{+} \subset \alpha_{\boldsymbol{\epsilon}}^{+}}} = \sum_{n_\infty = 0}^{s_{\boldsymbol{\epsilon}} + s_{\boldsymbol{\epsilon}}'} \sum_{\substack{\bar{\alpha}_{\boldsymbol{\epsilon}}^{-} \subset \alpha_{\boldsymbol{\epsilon}}^{-} \\ \bar{\alpha}_{\boldsymbol{\epsilon}}^{+} \subset \alpha_{\boldsymbol{\epsilon}}^{+} \\ \#\bar{\alpha}^- + \#\bar{\alpha}^+ = s_{\boldsymbol{\epsilon}} + s_{\boldsymbol{\epsilon}}' - n_\infty}} . \quad (108)$$

Each scalar products of separate states appearing in (97) ot (103) can now be expressed in terms of generalized Slavnov's determinants using the results of [113]. Using Theorem 3.3 of [113], we can write

$$\frac{\langle \bar{Q}_{\mathbf{A}_{m+1}} | Q_\tau \rangle}{\langle Q_\tau | Q_\tau \rangle} = 0 \qquad \text{if} \quad m < \#\bar{\alpha}_{\boldsymbol{\epsilon}}^- + \#\bar{\alpha}_{\boldsymbol{\epsilon}}^+, \quad (109)$$

$$= (-1)^{N(R - \bar{R})} 2^{R - \bar{R}} \frac{\prod_{j=1}^{\bar{R}} \left[ -a(\bar{q}_j) \prod_{k=1}^{R}(q_k - \bar{q}_j + \eta) \right]}{\prod_{j=1}^{R} \left[ -a(q_j) \prod_{k=1}^{R}(q_k - q_j + \eta) \right]} \frac{V(q_R, \ldots, q_1)}{V(\bar{q}_{\bar{R}}, \ldots, \bar{q}_1)}$$

$$\times \frac{\det_{\bar{R}} \mathcal{M}^{(-)}(\mathbf{q} | \bar{\mathbf{q}})}{\det_R \mathcal{N}^{(-)}(\mathbf{q})} \qquad \text{if} \quad m \ge \#\bar{\alpha}_{\boldsymbol{\epsilon}}^- + \#\bar{\alpha}_{\boldsymbol{\epsilon}}^+. \quad (110)$$

Here we have set

$$\bar{R} = \#\mathbf{A}_{m+1} = R + m - \#\bar{\alpha}_- - \#\bar{\alpha}_+, \tag{111}$$

$$\{q_j\}_{j \in \mathbf{A}_{m+1}} = \{\bar{q}_1, \dots, \bar{q}_{\bar{R}}\}. \tag{112}$$

Moreover, for $\bar{R} \geq R$, the matrix $\mathcal{M}^{(-)}(\mathbf{q} \,|\, \bar{\mathbf{q}})$ is defined in terms of the $R$-tuple $\mathbf{q} = (q_1, \dots, q_R)$ and of the $\bar{R}$-tuple $\bar{\mathbf{q}} = (\bar{q}_1, \dots, \bar{q}_{\bar{R}})$ as

$$\left[\mathcal{M}^{(-)}(\mathbf{q} \,|\, \bar{\mathbf{q}})\right]_{j,k} = \begin{cases} t(q_j - \bar{q}_k) + \mathfrak{a}_Q(\bar{q}_k)\, t(\bar{q}_k - q_j) & \text{if } j \leq R, \\ (\bar{q}_k)^{j-R-1} + \mathfrak{a}_Q(\bar{q}_k)\, (\bar{q}_k + \eta)^{j-R-1} & \text{if } j > R, \end{cases} \tag{113}$$

whereas the matrix $\mathcal{N}^{(-)}(\mathbf{q})$ is given by

$$\left[\mathcal{N}^{(-)}(\mathbf{q})\right]_{j,k} = \frac{\mathfrak{a}'_Q(q_j)}{\mathfrak{a}_Q(q_j)}\, \delta_{j,k} + K(q_j - q_k), \tag{114}$$

with $\mathfrak{a}_Q$ being given in terms of the roots $\{q_1, \dots, q_R\}$ of $Q \equiv Q_\tau$ as in (30) and

$$t(\lambda) = \frac{\eta}{\lambda(\lambda + \eta)}, \qquad K(\lambda) = t(\lambda) + t(-\lambda) = \frac{2\eta}{(\lambda + \eta)(\lambda - \eta)}. \tag{115}$$

Note that, for $\{q_1, \dots, q_R\}$ solution of the anti-periodic Bethe equations $\mathfrak{a}_Q(q_j) = 1$, $j = 1, \dots, R$, one has

$$\mathcal{M}^{(-)}(\mathbf{q} \,|\, \mathbf{q}) = \mathcal{N}^{(-)}(\mathbf{q}). \tag{116}$$

# 6 Infinite-size correlation functions of the anti-periodic XXX chain

We now explain how to take the thermodynamic limit of the result obtained in the previous section for $|Q_\tau\rangle$ being, in the homogeneous limit, one of the ground state of (1). This will lead to multiple integral representations for the zero-temperature correlation functions of the anti-periodic XXX chain in the thermodynamic limit which coincide in this limit with the results obtained in the periodic case in [17], and directly in the infinite size model in [24].

## 6.1 Vanishing and non-vanishing terms in the thermodynamic limit

In this subsection we find the conditions under which the terms of the expansion (103) are non-zero in the thermodynamic limit for $|Q_\tau\rangle$ being the ground state of the XXX chain (1).

We first compute the ratio of scalar products appearing in the last line of (103) in the thermodynamic limit.

**Proposition 6.1.** *Let $Q$ be a polynomial of the form*

$$Q(\lambda) = \prod_{j=1}^{R} (\lambda - q_j), \tag{117}$$

*with roots $q_1, \dots, q_R$ solving the system of anti-periodic Bethe equations $\mathfrak{a}_Q(q_j) = 1$, $j = 1, \dots, R$, where $\mathfrak{a}_Q$ is defined as in (30). We moreover suppose that $R$ scales as $N$ in the thermodynamic limit and that the roots $q_1, \dots, q_R$ become in these limits distributed on the real axis according to the density $\rho_{\text{tot}}$ (68), (57).*

*Let $\bar{Q}$ be a polynomial built from $Q$ in the form*

$$\bar{Q}(\lambda) = \prod_{j=1}^{R'}(\lambda - q_{\sigma_j})\prod_{k=1}^{m'}(\lambda - \xi_{\pi_k}), \tag{118}$$

*where $\sigma$ and $\pi$ are permutations of $\{1,\ldots,R\}$ and of $\{1,\ldots,N\}$ respectively, and where $R - R'$ and $m'$ remain finite in the thermodynamic limit.*

*Then,*

$$\frac{\langle Q|\bar{Q}\rangle}{\langle Q|Q\rangle} = \frac{\langle \bar{Q}|Q\rangle}{\langle Q|Q\rangle} = \begin{cases} 0 & \text{if } R' + m' < R, \\ o\left(\frac{1}{N^{R-R'}}\right) & \text{if } R' + m' > R, \end{cases} \tag{119}$$

*whereas, if $R' + m' = R$,*

$$\frac{\langle Q|\bar{Q}\rangle}{\langle Q|Q\rangle} = \frac{\langle \bar{Q}|Q\rangle}{\langle Q|Q\rangle} \underset{N \to \infty}{\sim} \prod_{j=1}^{m'}\left\{ \frac{a(\xi_{\pi_j})\prod_{k=1}^{R}(q_k - \xi_{\pi_j} + \eta)}{a(q_{\sigma_{R'+j}})\prod_{k=1}^{R}(q_k - q_{\sigma_{R'+j}} + \eta)} \prod_{i=1}^{R'}\frac{q_{\sigma_i} - q_{\sigma_{R'+j}}}{q_{\sigma_i} - \xi_{\pi_j}} \right\}$$
$$\times \prod_{1 \le i < j \le m'} \frac{q_{\sigma_{R'+i}} - q_{\sigma_{R'+j}}}{\xi_{\pi_i} - \xi_{\pi_j}} \det_{1 \le j,k \le m'} \frac{\rho(q_{\sigma_{R'+j}} - \xi_{\pi_k} + \eta/2)}{N \rho_{\text{tot}}(q_{\sigma_{R'+j}})}. \tag{120}$$

*Proof.* In the case $R' + m' < R$, it was shown in [113] that the ratio of scalar products vanishes (see (109)).

In the case $R' + m' \ge R$, the ratio of scalar products can be expressed from [113] as a ratio of determinants as in (110):

$$\frac{\langle Q|\bar{Q}\rangle}{\langle Q|Q\rangle} = \frac{\langle \bar{Q}|Q\rangle}{\langle Q|Q\rangle}$$

$$= (-1)^{N(R'+m'-R)}2^{R-R'-m'}\frac{\prod_{j=1}^{m'}\left[-a(\xi_{\pi_j})\prod_{k=1}^{R}(q_k - \xi_{\pi_j} + \eta)\right]}{\prod_{j=R'+1}^{R}\left[-a(q_{\sigma_j})\prod_{k=1}^{R}(q_k - q_{\sigma_j} + \eta)\right]}$$

$$\times \prod_{i=1}^{R'}\frac{\prod_{j=R'+1}^{R}(q_{\sigma_i} - q_{\sigma_j})}{\prod_{j=1}^{m'}(q_{\sigma_i} - \xi_{\pi_j})}\frac{\prod_{R'<i<j\le R}(q_{\sigma_i} - q_{\sigma_j})}{\prod_{1\le i<j\le m'}(\xi_{\pi_i} - \xi_{\pi_j})}\frac{\det_{R'+m'}\mathcal{M}^{(-)}(\mathbf{q}_\sigma|\bar{\mathbf{q}})}{\det_R\mathcal{N}^{(-)}(\mathbf{q}_\sigma)}, \tag{121}$$

in which we have used the notations of (113)-(114) and the shortcut notations $\mathbf{q}_\sigma = (q_{\sigma_1},\ldots,q_{\sigma_R})$ and $\bar{\mathbf{q}} = (q_{\sigma_1},\ldots,q_{\sigma_{R'}},\xi_{\pi_1},\ldots,\xi_{\pi_{m'}})$. More explicitly, $\mathcal{M}^{(-)}(\mathbf{q}_\sigma|\bar{\mathbf{q}})$ can be written as the following block matrix:

$$\mathcal{M}^{(-)}(\mathbf{q}_\sigma|\bar{\mathbf{q}}) = \begin{pmatrix} \mathcal{M}^{(1,1)} & \mathcal{M}^{(1,2)} \\ \mathcal{M}^{(2,1)} & \mathcal{M}^{(2,2)} \end{pmatrix}, \tag{122}$$

where $\mathcal{M}^{(1,1)}$, $\mathcal{M}^{(1,2)}$, $\mathcal{M}^{(2,1)}$ and $\mathcal{M}^{(2,2)}$ are respectively of size $R \times R'$, $R \times m'$, $\bar{n} \times R'$ and $\bar{n} \times m'$, with $\bar{n} = R' + m' - R$, with elements

$$\mathcal{M}^{(1,1)}_{j,k} = \mathcal{N}_{j,k}, \qquad\qquad j \le R, \; k \le R', \tag{123}$$

$$\mathcal{M}^{(1,2)}_{j,k} = t(q_{\sigma_j} - \xi_{\pi_k}), \qquad\qquad j \le R, \; k \le m', \tag{124}$$

$$\mathcal{M}^{(2,1)}_{j,k} = (q_{\sigma_k})^{j-1} + (q_{\sigma_k} + \eta)^{j-1}, \qquad j \le \bar{n}, \; k \le R', \tag{125}$$

$$\mathcal{M}^{(2,2)}_{j,k} = \xi_{\pi_k}^{j-1}, \qquad\qquad j \le \bar{n}, \; k \le m', \tag{126}$$

in which we have used the shortcut notation $\mathcal{N} = \mathcal{N}^{(-)}(\mathbf{q}_\sigma)$. Hence, the ratio of determinants in (121) can be written as

$$\frac{\det_{R'+m'}\mathcal{M}^{(-)}(\mathbf{q}_\sigma|\bar{\mathbf{q}})}{\det_R\mathcal{N}^{(-)}(\mathbf{q}_\sigma)} = \det_{R'+m'}\mathcal{S}, \tag{127}$$

where

$$S = \begin{pmatrix} \mathcal{N}^{-1}\mathcal{M}^{(1,1)} & \mathcal{N}^{-1}\mathcal{M}^{(1,2)} \\ \mathcal{M}^{(2,1)} & \mathcal{M}^{(2,2)} \end{pmatrix}, \tag{128}$$

with in particular $[\mathcal{N}^{-1}\mathcal{M}^{(1,1)}]_{j,k} = \delta_{j,k}$ for $j \le R, k \le R'$. The thermodynamic limit $N \to \infty$ of the matrix elements of $\mathcal{N}^{-1}\mathcal{M}^{(1,2)}$ can be computed similarly as in the periodic case [19] using the integral equation (56):

$$\left[\mathcal{N}^{-1}\mathcal{M}^{(1,2)}\right]_{j,k} = \frac{\rho(q_{\sigma_j} - \xi_{\pi_k} + \eta/2)}{N \rho_{\text{tot}}(q_{\sigma_j})} + o\left(\frac{1}{N}\right), \tag{129}$$

in which $\rho$ is given by (57) and $\rho_{\text{tot}}$ by (68). In particular, when $\bar{n} = R' + m' - R = 0$, we recover the result (120).

In the case $R' + m' > R$, it is convenient to rewrite $S$ (128) in terms of blocks of slightly different sizes:

$$S = \begin{pmatrix} \mathbb{I}_{R'} & \mathcal{S}^{(1,2)} \\ \mathcal{S}^{(2,1)} & \mathcal{S}^{(2,2)} \end{pmatrix}, \tag{130}$$

where $\mathbb{I}_{R'}$ is the identity square matrix of size $R'$, and where $\mathcal{S}^{(1,2)}$, $\mathcal{S}^{(2,1)}$ and $\mathcal{S}^{(2,2)}$ are respectively of size $R' \times m'$, $m' \times R'$ and $m' \times m'$, with elements

$$\mathcal{S}^{(1,2)}_{j,k} = \left[\mathcal{N}^{-1}\mathcal{M}^{(1,2)}\right]_{j,k} \tag{131}$$

$$\mathcal{S}^{(2,1)}_{j,k} = \begin{cases} 0 & \text{if } j \le R - R', \\ \mathcal{M}^{(2,1)}_{j-(R-R'),k} & \text{if } R - R' < j \le m', \end{cases} \tag{132}$$

$$\mathcal{S}^{(2,2)}_{j,k} = \begin{cases} \left[\mathcal{N}^{-1}\mathcal{M}^{(1,2)}\right]_{j+R',k} & \text{if } j \le R - R', \\ \mathcal{M}^{(2,2)}_{j-(R-R'),k} = \xi_{\pi_k}^{j-1-R+R'} & \text{if } R - R' < j \le m'. \end{cases} \tag{133}$$

Hence,

$$\det_{R'+m'} S = \det_{m'} S', \tag{134}$$

with $S' = \mathcal{S}^{(2,2)} - \mathcal{S}^{(2,1)}\mathcal{S}^{(1,2)}$, i.e.

$$\mathcal{S}'_{j,k} = \left[\mathcal{N}^{-1}\mathcal{M}^{(1,2)}\right]_{j+R',k} = \frac{\rho(q_{\sigma_{j+R'}} - \xi_{\pi_k} + \eta/2)}{N \rho_{\text{tot}}(q_{\sigma_{j+R'}})} + o\left(\frac{1}{N}\right) \quad \text{if } j \le R - R', \tag{135}$$

whereas, for $1 \le j \le m' + R' - R$,

$$\mathcal{S}'_{R-R'+j,k} = \mathcal{M}^{(2,2)}_{j,k} - \sum_{\ell=1}^{R'} \mathcal{M}^{(2,1)}_{j,\ell}\left[\mathcal{N}^{-1}\mathcal{M}^{(1,2)}\right]_{\ell,k}. \tag{136}$$

In particular, the $(R - R' + 1)$-th line of $S'$ is

$$\mathcal{S}'_{R-R'+1,k} = 1 - 2\sum_{\ell=1}^{R'}\left[\mathcal{N}^{-1}\mathcal{M}^{(1,2)}\right]_{\ell,k}$$

$$\xrightarrow[N \to \infty]{} 1 - 2\int_{-\infty}^{\infty} \rho(\lambda - \xi_{\pi_k} + \eta/2)\, d\lambda = 0, \tag{137}$$

which proves (119). $\qquad\qquad\square$

*Remark* 9. If we suppose moreover that the sums in (136) can be transformed into integrals $\forall j$, we obtain that all the lines of (136) vanish in the thermodynamic limit:

$$
\mathcal{S}'_{R-R'+j,k} \underset{N\to\infty}{\longrightarrow} \xi_{\pi_k}^{j-1} - \int_{-\infty}^{\infty} \left[ \lambda^{j-1} + (\lambda+\eta)^{j-1} \right] \rho(\lambda - \xi_{\pi_k} + \eta/2) \, d\lambda = 0 \,. \tag{138}
$$

Indeed, setting $\eta = -i$ and supposing that $|\Im(\xi_{\pi_k} + i/2)| < 1/2$, we have that

$$
\int_{-\infty}^{\infty} \lambda^{j-1} \rho(\lambda - \xi_{\pi_k} - i/2) \, d\lambda - \int_{-\infty}^{\infty} (\lambda-i)^{j-1} \rho(\lambda - \xi_{\pi_k} - i/2 - i) \, d\lambda
$$
$$
= -2\pi i \mathrm{Res}_{\lambda=\xi_{\pi_k}} \left[ \lambda^{j-1} \rho(\lambda - \xi_{\pi_k} - i/2) \right] = \xi_{\pi_k}^{j-1}, \quad \tag{139}
$$

and we can conclude by using the quasi-periodicity property $\rho(\lambda - i) = -\rho(\lambda)$. Note however that we do not need (138) for $j > 1$ for the proof of Proposition 6.1, it is enough that these lines remain finite in the thermodynamic limit.

As a consequence of this proposition, we can formulate the following corollary:

**Corollary 6.1.** *For a given $2m$-tuple $\epsilon \equiv (\epsilon_1, \ldots, \epsilon_{2m}) \in \{1,2\}^{2m}$, let us define the sets $\alpha_\epsilon^-$ and $\alpha_\epsilon^+$ of respective cardinality $s_\epsilon$ and $s_\epsilon'$ as in (102)- (101), and let us consider the matrix element $F_m(\tau, \epsilon)$ in a state $|Q_\tau\rangle$ with $Q_\tau \equiv Q$ satisfying the same hypothesis as in Proposition 6.1. Then*

$$
\lim_{N\to\infty} F_m(\tau, \epsilon) = 0 \qquad if \quad s_\epsilon + s_\epsilon' \neq m \,. \tag{140}
$$

*Moreover, if $s_\epsilon + s_\epsilon' = m$, the non-vanishing contribution of $F_m(\tau, \epsilon)$ in the thermodynamic limit is given by*

$$
\lim_{N\to\infty} F_m(\tau, \epsilon) = \lim_{N\to\infty} \prod_{k=1}^{m} \frac{1}{\tau(\xi_k)} \sum_{\{a_j, a_j'\}} \prod_{j\in\alpha_\epsilon^-} \left( \frac{d(q_{a_j}) \prod_{\substack{k=1 \\ k\in\mathbf{A}_j}}^{R+j-1} (q_{a_j} - q_k + \eta)}{\prod_{\substack{k=1 \\ k\in\mathbf{A}_j'}}^{R+j} (q_{a_j} - q_k)} \right)
$$
$$
\times \prod_{j\in\alpha_\epsilon^+} \left( -\frac{a(q_{a_j'}) \prod_{\substack{k=1 \\ k\in\mathbf{A}_j'}}^{R+j-1} (q_k - q_{a_j'} + \eta)}{\prod_{\substack{k=1 \\ k\in\mathbf{A}_{j+1}}}^{R+j} (q_k - q_{a_j'})} \right) \frac{\langle \bar{Q}_{\mathbf{A}_{m+1}} | Q_\tau \rangle}{\langle Q_\tau | Q_\tau \rangle}, \quad \tag{141}
$$

*in which the summation is taken over the indices $a_j$ for $j \in \alpha_\epsilon^-$ and $a_j'$ for $j \in \alpha_\epsilon^+$ satisfying (104)-(106), and where we have used the notation (107).*

In other words, it means that, in the thermodynamic limit, we recover the same selection rules (140) for the elementary blocks as in the periodic case. Moreover, the only non-vanishing terms in the series (103) corresponds to $\bar{\alpha}_\epsilon^+ = \alpha_\epsilon^+$ and $\bar{\alpha}_\epsilon^- = \alpha_\epsilon^-$, i.e. to $n_\infty = 0$. This means that the residues of the poles at infinity that appeared when moving the integration contours in the computation of the action of Section 5.1 (see Proposition 5.2 and Corollary 5.1) do not contribute to the thermodynamic limit of the correlation functions.

*Proof.* Let us consider the expansion (103) for $F_m(\tau, \epsilon)$, which involves multiple sums over indices $\{a_j, a_j'\}$.

For a given term of the sum, the polynomial $\bar{Q}_{\mathbf{A}_{m+1}}$ is of the form (118) with $R - R'$ equal to the number of indices $a_j$ or $a_j'$ in the multiple sums which are taken between 1 and $R$. On the other hand, each of the sums over an index $a_j$ or $a_j'$ from 1 to $R$ leads to an integral in the thermodynamic limit provided it is balanced by a factor $1/N$, the other terms of the sums (for

$a_j$ or $a'_j$ from $R+1$ to $R+m$) contributing to order 1 to the thermodynamic limit. Hence, the non-vanishing contributions in the thermodynamic limits correspond to the configurations in the expansion (103) for which the ratio of determinants is exactly of order $O(1/N^{R-R'})$, which, from Proposition 6.1, happens only when the two polynomials $\bar{Q}_{\mathbf{A}_{m+1}}$ and $Q_\tau$ are of the same degree $R$, i.e. when $\#\bar{\alpha}^-_\epsilon + \#\bar{\alpha}^+_\epsilon = m$.

Since $\#\bar{\alpha}^-_\epsilon + \#\bar{\alpha}^+_\epsilon \leq \#\alpha^-_\epsilon + \#\alpha^+_\epsilon = s_\epsilon + s'_\epsilon$, the whole sum (103) is vanishing in the thermodynamic limit if $s_\epsilon + s'_\epsilon < m$, so that

$$\lim_{N\to\infty} F_m(\tau,\epsilon) = 0 \qquad \text{if} \quad s_\epsilon + s'_\epsilon < m. \tag{142}$$

If instead $s_\epsilon + s'_\epsilon > m$ we use the symmetry (75) and the fact that $s_{3-\epsilon} + s'_{3-\epsilon} < m$ to conclude that

$$\lim_{N\to\infty} F_m(\tau,\epsilon) = \lim_{N\to\infty} F_m(\tau, 3-\epsilon) = 0 \qquad \text{if} \quad s_\epsilon + s'_\epsilon > m. \tag{143}$$

This proves (140).

If now $s_\epsilon + s'_\epsilon = m$, the only terms contributing to the thermodynamic limit of $F_m(\tau,\epsilon)$ in the sum (103) are those for which $\#\bar{\alpha}^-_\epsilon + \#\bar{\alpha}^+_\epsilon = \#\alpha^-_\epsilon + \#\alpha^+_\epsilon$, i.e. $\bar{\alpha}^\pm_\epsilon = \alpha^\pm_\epsilon$, which also proves (141). $\qquad\square$

Note that, by using the explicit expression for the transfer matrix eigenvalue evaluated at $\xi_k$, $k = 1, \ldots, m$,

$$\tau(\xi_k) = -a(\xi_k) \frac{Q_\tau(\xi_k - \eta)}{Q_\tau(\xi_k)}, \tag{144}$$

together with the Bethe equations

$$d(q_{a_j}) = a(q_{a_j}) \frac{Q_\tau(q_{a_j} - \eta)}{Q_\tau(q_{a_j} + \eta)}, \qquad \forall\, a_j \leq R, \tag{145}$$

and the observation that $d(q_{a_j}) = 0$ for any $a_j > R$, one can rewrite (141) in the following way

$$\lim_{N\to\infty} F_m(\tau,\epsilon) = \lim_{N\to\infty} \prod_{k=1}^m \frac{Q_\tau(\xi_k)}{a(\xi_k) Q_\tau(\xi_k - \eta)}$$
$$\times \sum_{\{a_j, a'_j\}} \prod_{j \in \alpha^-_\epsilon} \left( -a(q_{a_j}) \frac{Q_\tau(q_{a_j} - \eta)}{Q_\tau(q_{a_j} + \eta)} \frac{\prod_{\substack{k=1 \\ k \in \mathbf{A}_j}}^{R+j-1}(q_{a_j} - q_k + \eta)}{\prod_{\substack{k=1 \\ k \in \mathbf{A}'_j}}^{R+j}(q_{a_j} - q_k)} \right)$$
$$\times \prod_{j \in \alpha^+_\epsilon} \left( a(q_{a'_j}) \frac{\prod_{\substack{k=1 \\ k \in \mathbf{A}'_j}}^{R+j-1}(q_k - q_{a'_j} + \eta)}{\prod_{\substack{k=1 \\ k \in \mathbf{A}_{j+1}}}^{R+j}(q_k - q_{a'_j})} \right) \frac{\langle \bar{Q}_{\mathbf{A}_{m+1}} | Q_\tau \rangle}{\langle Q_\tau | Q_\tau \rangle}, \tag{146}$$

where the summation is taken here over the indices $a_j$ for $j \in \alpha^-_\epsilon$ and $a'_j$ for $j \in \alpha^+_\epsilon$ such that

$$1 \leq a_j \leq R, \quad a_j \in \mathbf{A}_j, \qquad 1 \leq a'_j \leq R+j, \quad a'_j \in \mathbf{A}'_j. \tag{147}$$

## 6.2 Multiple integral representation for the correlation functions in the thermodynamic limit

Let us now now consider, for any $2m$-tuple $\epsilon \equiv (\epsilon_1, \ldots, \epsilon_{2m})$, the matrix elements

$$\mathcal{F}_m(\epsilon) = \lim_{N\to\infty} \frac{\langle Q_\tau | \prod_{j=1}^m E_j^{\epsilon_{2j-1}, \epsilon_{2j}} | Q_\tau \rangle}{\langle Q_\tau | Q_\tau \rangle}, \tag{148}$$

for $|Q_\tau\rangle$ being an eigenstate of the transfer matrix (5) described in the thermodynamic limit by the density of roots $\rho_{\text{tot}}$, and which tends to one of the ground states of the anti-periodic XXX chain (1) in the homogeneous limit. It follows from Corollary 6.1, (120) and (146) that the terms contributing to the thermodynamic limit in the anti-periodic model are exactly of the same form that the terms contributing to the thermodynamic limit in the periodic case, see formulas (4.6)-(4.7) and (5.3)-(5.4) (in which we use the periodic analog of (144) and (145)) of [17]. Hence their thermodynamic limit coincide.

Therefore we obtain the following multiple integral representation for the correlation functions (148) in the thermodynamic limit, which coincides with the results of [17, 24]:

$$
\mathcal{F}_m(\epsilon) = \delta_{s_\epsilon + s'_\epsilon, m} \prod_{k<l} \frac{\sinh \pi(\xi_k - \xi_l)}{\xi_k - \xi_l} \prod_{j=1}^{s'_\epsilon} \int_{-\infty-i}^{\infty-i} \frac{d\lambda_j}{2i} \prod_{j=s'_\epsilon+1}^{m} \int_{-\infty}^{\infty} i\frac{d\lambda_j}{2} \prod_{a>b} \frac{\sinh \pi(\lambda_a - \lambda_b)}{\lambda_a - \lambda_b - i}
$$

$$
\times \prod_{a=1}^{m}\prod_{k=1}^{m} \frac{1}{\sinh \pi(\lambda_a - \xi_k)} \prod_{j\in\alpha_\epsilon^-} \left[ \prod_{k=1}^{j-1}(\mu_j - \xi_k - i) \prod_{k=j+1}^{m}(\mu_j - \xi_k) \right]
$$

$$
\times \prod_{j\in\alpha_\epsilon^+} \left[ \prod_{k=1}^{j-1}(\mu'_j - \xi_k + i) \prod_{k=j+1}^{m}(\mu'_j - \xi_k) \right], \quad (149)
$$

in which the sets $\alpha_\epsilon^-$ and $\alpha_\epsilon^+$ are defined as in (101)-(102), and the integration parameters are ordered as

$$
(\lambda_1, \ldots, \lambda_m) = (\mu'_{j'_{\max}}, \ldots, \mu'_{j'_{\min}}, \mu_{j_{\min}}, \ldots, \mu_{j_{\max}}), \quad (150)
$$

with

$$
j'_{\min} = \min\{j \,|\, j \in \alpha_\epsilon^+\}, \qquad j'_{\max} = \max\{j \,|\, j \in \alpha_\epsilon^+\}, \quad (151)
$$

$$
j_{\min} = \min\{j \,|\, j \in \alpha_\epsilon^-\}, \qquad j_{\max} = \max\{j \,|\, j \in \alpha_\epsilon^-\}. \quad (152)
$$

In the homogeneous limit ($\xi_j = -i/2, \forall j$) the correlation function $\mathcal{F}_m(\epsilon)$ has the following form:

$$
\mathcal{F}_m(\epsilon) = \delta_{s_\epsilon + s'_\epsilon, m} (-1)^{s_\epsilon} (-\pi)^{\frac{m(m+1)}{2}} \prod_{j=1}^{s'_\epsilon} \int_{-\infty-i}^{\infty-i} \frac{d\lambda_j}{2\pi} \prod_{j=s'_\epsilon+1}^{m} \int_{-\infty}^{\infty} \frac{d\lambda_j}{2\pi} \prod_{a>b} \frac{\sinh \pi(\lambda_a - \lambda_b)}{\lambda_a - \lambda_b - i}
$$

$$
\times \prod_{j\in\alpha_\epsilon^-} \frac{(\mu_j - \frac{i}{2})^{j-1} (\mu_j + \frac{i}{2})^{m-j}}{\cosh^m(\pi\mu_j)} \prod_{j\in\alpha_\epsilon^+} \frac{(\mu'_j + \frac{3i}{2})^{j-1} (\mu'_j + \frac{i}{2})^{m-j}}{\cosh^m(\pi\mu'_j)}. \quad (153)
$$

## 7 Correlation functions of the XXX chain with a non-diagonal twist

In the previous sections, we have shown how to compute the (elementary building blocks of the) correlation functions in the XXX chain with anti-periodic boundary conditions. It is interesting to see how the method and results presented above are modified in the case of a chain with a more general non-diagonal twist. This is the purpose of this section.

Let us consider a generic invertible $2 \times 2$ matrix,

$$
K = \begin{pmatrix} a & b \\ c & d \end{pmatrix}, \quad (154)
$$

and let us define the monodromy matrix with twist $K$ as

$$T_0^{(K)}(\lambda) = K_0 T_0(\lambda)$$
$$= \begin{pmatrix} A^{(K)}(\lambda) = \mathsf{a}A(\lambda) + \mathsf{b}C(\lambda) & B^{(K)}(\lambda) = \mathsf{a}B(\lambda) + \mathsf{b}D(\lambda) \\ C^{(K)}(\lambda) = \mathsf{c}A(\lambda) + \mathsf{d}C(\lambda) & D^{(K)}(\lambda) = \mathsf{c}B(\lambda) + \mathsf{d}D(\lambda) \end{pmatrix}, \tag{155}$$

to which is associated the one-parameter family of commuting transfer matrices:

$$\mathcal{T}^{(K)}(\lambda) = \mathrm{tr}_0 \, T_0^{(K)}(\lambda). \tag{156}$$

## 7.1 Diagonalisation of the transfer matrix by the SoV method

Under the condition $\mathsf{b} \neq 0$, one can apply Sklyanin's SoV approach [3,4], see also [114]. Here, we follow the presentation given in section 2 of [117] and in [127] for the diagonalization of the transfer matrix in this general twisted case.

The separate variables are generated by the operator zeros of $B^{(K)}(\lambda)$. The latter is diagonalizable with simple spectrum,

$$_K\langle \mathbf{h} | B^{(K)}(\lambda) = \mathsf{b} \prod_{n=1}^{N} (\lambda - \xi_n + h_n \eta)\,_K\langle \mathbf{h} |,$$
$$= \mathsf{b}\, d_{\mathbf{h}}(\lambda)\,_K\langle \mathbf{h} |, \qquad \forall \mathbf{h} \equiv (h_1, \dots, h_N) \in \{0, 1\}^{\otimes N}, \tag{157}$$

and the elements $_K\langle \mathbf{h} |$ of the corresponding SoV eigenbasis can be constructed as

$$_K\langle \mathbf{h} | \equiv \langle 0 | \prod_{n=1}^{N} \left( \frac{A^{(K)}(\xi_n)}{\mathsf{k}_1\, d(\xi_n - \eta)} \right)^{h_n}, \qquad \forall \mathbf{h} \equiv (h_1, \dots, h_N) \in \{0, 1\}^{\otimes N}, \tag{158}$$

where we have defined $\langle 0 | = \bigotimes_{n=1}^{N} (1, 0)_n$. For convenience, we choose the normalization coefficient $\mathsf{k}_1$ in (158) such that

$$\mathsf{k}_1^2 - \mathsf{k}_1 \, \mathrm{tr} K + \det K = 0, \tag{159}$$

i.e. $\mathsf{k}_1$ is an eigenvalue of the matrix $K$. Setting

$$\mathsf{k}_2 = \frac{\det K}{\mathsf{k}_1}, \tag{160}$$

i.e. $\mathsf{k}_2$ is the second eigenvalue of $K$, we can compute the SoV action of the remaining Yang-Baxter generators on the basis (158) as

$$_K\langle \mathbf{h} | A^{(K)}(\lambda) = \mathsf{a}\, d_{\mathbf{h}}(\lambda)\,_K\langle \mathbf{h} | + \mathsf{k}_1 \sum_{n=1}^{N} \delta_{h_a,0}\, d(\xi_a^{(1)}) \prod_{b \neq a} \frac{\lambda - \xi_b^{(h_b)}}{\xi_a^{(h_a)} - \xi_b^{(h_b)}}\,_K\langle T_a^+ \mathbf{h} |, \tag{161}$$

$$_K\langle \mathbf{h} | D^{(K)}(\lambda) = \mathsf{d}\, d_{\mathbf{h}}(\lambda)\,_K\langle \mathbf{h} | + \mathsf{k}_2 \sum_{a=1}^{N} \delta_{h_a,1}\, a(\xi_a^{(0)}) \prod_{b \neq a} \frac{\lambda - \xi_b^{(h_b)}}{\xi_a^{(h_a)} - \xi_b^{(h_b)}}\,_K\langle T_a^- \mathbf{h} |, \tag{162}$$

while the SoV representation of $C^{(K)}(\lambda)$ follows from the above ones and the quantum determinant condition.

Similarly, following Corollary B.2 of [127], the right SoV basis of $\mathcal{H}$ can be constructed as

$$| \mathbf{h} \rangle_K \equiv \frac{1}{\mathsf{n}} \prod_{n=1}^{N} \left( \frac{A^{(K)}(\xi_n - \eta)}{\mathsf{k}_1\, d(\xi_n - \eta)} \right)^{1 - h_n} | \underline{0} \rangle, \qquad \forall \mathbf{h} \equiv (h_1, \dots, h_N) \in \{0, 1\}^{\otimes N}, \tag{163}$$

where $|\underline{0}\rangle = \bigotimes_{n=1}^{N} \binom{0}{1}_n$ and n is a normalization coefficient. Then, the SoV action of the Yang-Baxter generators on (163) is

$$B^{(K)}(\lambda)|\mathbf{h}\rangle_K = \mathsf{b}\, d_{\mathbf{h}}(\lambda)|\mathbf{h}\rangle_K, \tag{164}$$

$$A^{(K)}(\lambda)|\mathbf{h}\rangle_K = \mathsf{a}\, d_{\mathbf{h}}(\lambda)|\mathbf{h}\rangle_K + \mathsf{k}_1 \sum_{a=1}^{N} \delta_{h_a,1}\, d(\xi_a^{(1)}) \prod_{b\neq a} \frac{\lambda - \xi_b^{(h_b)}}{\xi_a^{(h_a)} - \xi_b^{(h_b)}}|\mathsf{T}_a^-\mathbf{h}\rangle_K, \tag{165}$$

$$D^{(K)}(\lambda)|\mathbf{h}\rangle_K = \mathsf{d}\, d_{\mathbf{h}}(\lambda)|\mathbf{h}\rangle_K + \mathsf{k}_2 \sum_{a=1}^{N} \delta_{h_a,0}\, a(\xi_a^{(0)}) \prod_{b\neq a} \frac{\lambda - \xi_b^{(h_b)}}{\xi_a^{(h_a)} - \xi_b^{(h_b)}}|\mathsf{T}_a^+\mathbf{h}\rangle_K, \tag{166}$$

and, with an adequate choice of the normalization coefficient n, it holds:

$$_K\langle \mathbf{h}|\mathbf{k}\rangle_K = \frac{\delta_{\mathbf{h},\mathbf{k}}}{V(\xi_1^{(h_1)},\dots,\xi_N^{(h_N)})}. \tag{167}$$

Note moreover that, as proven in [117], the transfer matrix is diagonalizable and with simple spectrum as soon as the same properties holds for the twist matrix $K$. In the SoV bases, the eigencovector of the transfer matrix can be written in the form

$$_K\langle Q_\tau| = \sum_{\mathbf{h}\in\{0,1\}^N} \prod_{n=1}^{N} Q_\tau(\xi_n^{(h_n)})\, V(\xi_1^{(h_1)},\dots,\xi_N^{(h_N)})\, _K\langle \mathbf{h}|, \tag{168}$$

and the eigenvector has the form

$$|Q_\tau\rangle_K = \sum_{\mathbf{h}\in\{0,1\}^N} \prod_{n=1}^{N} \left\{ \left(-\frac{\mathsf{k}_2}{\mathsf{k}_1}\right)^{h_n} Q_\tau(\xi_n^{(h_n)}) \right\} V(\xi_1^{(1-h_1)},\dots,\xi_N^{(1-h_N)})|\mathbf{h}\rangle_K, \tag{169}$$

where $Q_\tau(\lambda)$ is a polynomial of degree $R \leq N$ satisfying with the corresponding transfer matrix eigenvalue $\tau(\lambda)$ the following TQ-equation (see Theorem 3.2 of [117]):

$$\tau(\lambda)Q_\tau(\lambda) = \mathsf{k}_2\, a(\lambda)Q_\tau(\lambda - \eta) + \mathsf{k}_1\, d(\lambda)Q_\tau(\lambda + \eta). \tag{170}$$

The same construction (158)-(170) can be done by exchanging the role of $\mathsf{k}_1$ and $\mathsf{k}_2$, and the eigenstates of (156) can alternatively be constructed in terms of a polynomial $\widehat{Q}_\tau(\lambda)$ of degree $S \leq N$ solving with $\tau(\lambda)$ the second TQ-equation

$$\tau(\lambda)\widehat{Q}_\tau(\lambda) = \mathsf{k}_1\, a(\lambda)\widehat{Q}_\tau(\lambda - \eta) + \mathsf{k}_2\, d(\lambda)\widehat{Q}_\tau(\lambda + \eta). \tag{171}$$

The two polynomials $Q_\tau$ and $\widehat{Q}_\tau$ then satisfy the quantum Wronskian relation

$$\mathsf{k}_2\, \widehat{Q}_\tau(\lambda)Q_\tau(\lambda - \eta) - \mathsf{k}_1\, Q_\tau(\lambda)\widehat{Q}_\tau(\lambda - \eta) = (\mathsf{k}_2 - \mathsf{k}_1)\, d(\lambda), \tag{172}$$

implying in particular that $R + S = N$.

As in the anti-periodic case (31), (32), (36), (38), the transfer matrix eigenstates can be written in the form of generalized Bethe states in terms of the roots $q_1,\dots,q_R$ of $Q_\tau(\lambda)$,

$$_K\langle Q_\tau| \propto\, _K\langle 1|\prod_{k=1}^{R} B^{(K)}(q_k), \qquad |Q_\tau\rangle_K \propto \prod_{k=1}^{R} B^{(K)}(q_k)|1\rangle_K, \tag{173}$$

where

$$_K\langle 1| = \sum_{\mathbf{h}} V(\xi_1^{(h_1)},\dots,\xi_N^{(h_N)})\, _K\langle \mathbf{h}|, \tag{174}$$

$$|1\rangle_K = \sum_{\mathbf{h}} \prod_{n=1}^{N} \left(-\frac{\mathsf{k}_2}{\mathsf{k}_1}\right)^{h_n} V(\xi_1^{(1-h_1)},\dots,\xi_N^{(1-h_N)})|\mathbf{h}\rangle_K, \tag{175}$$

are eigenstates of the transfer matrix with eigenvalue $k_2\, a(\lambda) + k_1\, d(\lambda)$, or in terms of the roots $\widehat{q}_1, \ldots, \widehat{q}_{N-R}$ of $\widehat{Q}_\tau(\lambda)$,

$$_K\langle Q_\tau | \propto\, _K\langle 1_{\text{alt}} | \prod_{k=1}^{N-R} B^{(K)}(\widehat{q}_k), \qquad | Q_\tau \rangle_K \propto \prod_{k=1}^{N-R} B^{(K)}(\widehat{q}_k) | 1_{\text{alt}} \rangle_K, \tag{176}$$

where

$$_K\langle 1_{\text{alt}} | = \sum_{\mathbf{h}} \prod_{n=1}^{N} \left( \frac{k_1}{k_2} \right)^{h_n} V(\xi_1^{(h_1)}, \ldots, \xi_N^{(h_N)})\, _K\langle \mathbf{h} |, \tag{177}$$

$$| 1_{\text{alt}} \rangle_K = \sum_{\mathbf{h}} \prod_{n=1}^{N} (-1)^{h_n}\, V(\xi_1^{(1-h_1)}, \ldots, \xi_N^{(1-h_N)}) | \mathbf{h} \rangle_K, \tag{178}$$

are eigenstates of the transfer matrix with eigenvalue $k_1\, a(\lambda) + k_2\, d(\lambda)$.

*Remark* 10. In the triangular case $c = 0$ with $a = k_1$ and $d = k_2$ ($b \neq 0$, $k_1 \neq k_2$), the eigen-covector (168) and eigenvector (169) of the transfer matrix can be written as the following usual Bethe states:

$$_K\langle Q_\tau | \propto \langle \underline{0} | \prod_{k=1}^{R} B^{(K)}(q_k) \quad \in \mathcal{H}_{-N/2,\ldots,R-N/2}^*, \tag{179}$$

$$| Q_\tau \rangle_K \propto \prod_{k=1}^{N-R} B^{(K)}(\widehat{q}_k) | 0 \rangle \quad \in \mathcal{H}_{R-N/2,\ldots,N/2}, \tag{180}$$

where we have defined

$$\mathcal{H}_{-N/2,\ldots,S-N/2} = \bigoplus_{n=-N/2,1-N/2,\ldots,S-N/2} \mathcal{H}_n, \tag{181}$$

$$\mathcal{H}_{S-N/2,\ldots,N/2} = \bigoplus_{n=S-N/2,S+1-N/2,\ldots,N/2} \mathcal{H}_n, \tag{182}$$

with $\mathcal{H}_n$ being the $S_z$-eigenspace associated to the eigenvalue $n$. Indeed, it is easy to see that $|0\rangle$ and $\langle \underline{0} |$ are transfer matrix eigenstates with respective eigenvalues $k_1\, a(\lambda) + k_2\, d(\lambda)$ and $k_2\, a(\lambda) + k_1\, d(\lambda)$, and therefore the simplicity of the spectrum implies that $|0\rangle \propto | 1_{\text{alt}} \rangle_K$ and $\langle \underline{0} | \propto\, _K\langle 1 |$. Note that such eigenstates could have been directly constructed within ABA, but in the latter framework the description is only partial: an ABA construction of the eigenvector and eigencovector in terms of the *same* set of roots is actually missing, which makes uneasy the computation of the scalar products. Instead, the SoV construction provides us with a full description and the scalar products can be computed as in [113]. The triangular case $b = 0$ with $c \neq 0$ can be treated similarly, by exchanging the SoV construction w.r.t. $B^{(K)}(\lambda)$ with the SoV construction w.r.t. $C^{(K)}(\lambda)$.

The solutions of the Bethe equations following from (170), and in particular the ground state, can be studied as in section 4. We shall restrict our study to twists $K$ satisfying the physical constraint[14]

$$\frac{k_2}{k_1} = e^{i\pi\alpha} \qquad \text{with} \quad -1 < \alpha \leq 1. \tag{183}$$

Then the Bethe equations can be written in logarithmic form as

$$\widehat{\xi}_Q(\lambda_j) = \frac{2n_j - N + R - 1 + \alpha}{N}\, \pi, \qquad n_j \in \mathbb{Z}, \tag{184}$$

---

[14]For a physical model with a Hermitian Hamiltonian, the matrix $K$ is unitary and the ratio of its eigenvalues obviously satisfies (183).

where $\widehat{\xi}_Q$ is still given by (51). Hence, there is simply a shift on the real axis with respect to the known periodic case (corresponding to $\alpha = 0$) or with respect to the anti-periodic (or also the $\sigma^z$-twisted) case (corresponding to $\alpha = 1$), and the density of Bethe roots for the ground state on the real axis remains the same (57).

## 7.2 Action on a separate state

It is possible to compute the action of products of local operators on a transfer matrix eigenstate by proceeding as in the anti-periodic case.

We have the following reconstruction, which is the analog of Proposition 5.1, in terms of the twisted transfer matrix (156):

**Proposition 7.1.** *Let K be an invertible $2 \times 2$ matrix, and let $X_n \in \text{End}(V_n)$. Then*

$$X_n = \prod_{k=1}^{n-1} \mathcal{T}^{(K)}(\xi_k) \, \text{tr}_0 \Big[ X_0 \, T_0^{(K)}(\xi_n) \Big] \prod_{k=1}^{n} \big[ \mathcal{T}^{(K)}(\xi_k) \big]^{-1} , \tag{185}$$

$$= \prod_{b=1}^{n} \mathcal{T}^{(K)}(\xi_b) \frac{\text{tr}_0 \Big[ \tilde{X}_0 \, T_0^{(K)}(\xi_n - \eta) \Big]}{a(\xi_n) \, d(\xi_n - \eta) \, \det K} \prod_{b=1}^{n-1} \big[ \mathcal{T}^{(K)}(\xi_k) \big]^{-1} , \tag{186}$$

*where $\tilde{X}$ denotes the adjoint matrix of the matrix X, i.e.*

$$\tilde{X}X = X\tilde{X} = \det X \, \text{Id} . \tag{187}$$

*Proof.* See [111], in which a direct proof is given in the more complicated dynamical 6-vertex case. In this simpler twisted XXX case, it is also possible to propose an alternative proof based on the known reconstructions in the periodic case [17]:

$$X_n = \prod_{b=1}^{n-1} \mathcal{T}^{(I)}(\xi_b) \, \text{tr}_0 \Big[ X_0 \, T_0(\xi_n) \Big] \prod_{b=1}^{n} \big[ \mathcal{T}^{(I)}(\xi_b) \big]^{-1} , \tag{188}$$

$$= \prod_{b=1}^{n} \mathcal{T}^{(I)}(\xi_b) \frac{\text{tr}_0 \Big[ \tilde{X}_0 \, T_0(\xi_n - \eta) \Big]}{a(\xi_n) \, d(\xi_n - \eta)} \prod_{b=1}^{n-1} \big[ \mathcal{T}^{(I)}(\xi_b) \big]^{-1} , \tag{189}$$

with

$$\tilde{X} = \sigma^y \, X^t \, \sigma^y . \tag{190}$$

By using these results we can write

$$K_m = \prod_{b=1}^{m-1} \mathcal{T}^{(I)}(\xi_b) \, \mathcal{T}^{(K)}(\xi_m) \prod_{b=1}^{m} \big[ \mathcal{T}^{(I)}(\xi_b) \big]^{-1} , \tag{191}$$

$$\tilde{K}_m = \prod_{b=1}^{m} \mathcal{T}^{(I)}(\xi_b) \frac{\mathcal{T}^{(K)}(\xi_m - \eta)}{a(\xi_m) \, d(\xi_m - \eta)} \prod_{b=1}^{m-1} \big[ \mathcal{T}^{(I)}(\xi_b) \big]^{-1} , \tag{192}$$

and so

$$\mathcal{T}^{(K)}(\xi_m) \frac{\mathcal{T}^{(K)}(\xi_m - \eta)}{a(\xi_m) \, d(\xi_m - \eta)} = \frac{\mathcal{T}^{(K)}(\xi_m - \eta)}{a(\xi_m) \, d(\xi_m - \eta)} \mathcal{T}^{(K)}(\xi_m) = \det K \, \text{Id} , \tag{193}$$

$$\prod_{m=1}^{r} K_m = \prod_{b=1}^{r} \mathcal{T}^{(K)}(\xi_b) \prod_{b=1}^{r} \big[ \mathcal{T}^{(I)}(\xi_b) \big]^{-1} , \tag{194}$$

$$\prod_{m=1}^{s} \tilde{K}_m = \prod_{b=1}^{s} \mathcal{T}^{(I)}(\xi_b) \prod_{b=1}^{s} \frac{\mathcal{T}^{(K)}(\xi_m^{(1)})}{a(\xi_m) \, d(\xi_m - \eta)} . \tag{195}$$

From the identity

$$X_n = \left(\prod_{m=1}^{n-1} K_m\right) Y_n \left(\prod_{m=1}^{n} K_m^{-1}\right), \tag{196}$$

with $Y_n = X_n K_n$, we now obtain (185) by expressing $Y_n$ using the periodic reconstruction (188) and the product of $K$ and $K^{-1}$ by (194) and (195) respectively.

From the identity

$$X_n = \left(\prod_{m=1}^{n} K_m\right) Z_n \left(\prod_{m=1}^{n-1} K_m^{-1}\right), \tag{197}$$

with $Z_n = K_n^{-1} X_n$, we obtain (186) by expressing $Z_n$ using the periodic reconstruction (189) and the product of $K$ and $K^{-1}$ by (194) and (195) respectively. □

For any $\lambda \notin \{\xi_i - \eta, \xi_i - 2\eta \mid i = 1,\ldots,N\}$, we can then define, similarly as in (77), the operators:

$$\bar{T}_{\epsilon,\epsilon'}^{(K)}(\lambda) = \begin{cases} \left[B^{(K)}(\lambda+\eta)\right]^{-1} A^{(K)}(\lambda+\eta) D^{(K)}(\lambda) & \text{if } (\epsilon,\epsilon') = (2,1), \\ T_{\epsilon,\epsilon'}^{(K)}(\lambda) & \text{otherwise,} \end{cases} \tag{198}$$

since $B^{(K)}(\lambda)$ is invertible for any $\lambda \neq \xi_i, \xi_i - \eta, i = 1,\ldots N$. The condition $\det_q T^{(K)}(\xi_i+\eta) = 0$, then implies the identities:

$$\bar{T}_{\epsilon,\epsilon'}^{(K)}(\xi_i) = T_{\epsilon,\epsilon'}^{(K)}(\xi_i) \qquad \forall i \in \{1,\ldots,N\}, \quad \forall \epsilon,\epsilon' \in \{1,2\}, \tag{199}$$

so that the reconstruction of local operators (185) can be written in terms of the matrix elements $\bar{T}_{\epsilon,\epsilon'}^{(K)}$ instead of $T_{\epsilon,\epsilon'}^{(K)}$.

The action of the operators (198) on a separate state of the form

$$_K\langle Q| = \sum_{\mathbf{h}\in\{0,1\}^N} \prod_{n=1}^{N} Q(\xi_n^{(h_n)})\, V(\xi_1^{(h_1)},\ldots,\xi_N^{(h_N)})\, _K\langle \mathbf{h}|, \tag{200}$$

where $Q(\lambda) = \prod_{k=1}^{R}(\lambda - q_k)$ is a polynomial of degree $R \leq N$, can easily be computed in terms of multiple contour integrals, as in Proposition 5.1. More precisely, the analog of Proposition 5.1 in the case of the twist $K$ (154) can be formulated as follows:

**Proposition 7.1.** *Let $\lambda$ be a generic parameter. The left action of the operator $\bar{T}_{\epsilon_2,\epsilon_1}^{(K)}(\lambda)$, $\epsilon_1,\epsilon_2 \in \{1,2\}$, on a generic separate state $_K\langle Q|$ of the form (200) can be written as the following sum of contour integrals:*

$$_K\langle Q|\, \bar{T}_{\epsilon_2,\epsilon_1}^{(K)}(\lambda) = \sum_{\mathbf{h}} \mathsf{b}\, d_{\mathbf{h}}(\lambda) \prod_{n=1}^{N} Q(\xi_n^{(h_n)})$$

$$\times \left[\left(\frac{\mathsf{d}}{\mathsf{b}}\oint_{\mathcal{C}_\infty} + \frac{\mathsf{k}_2}{\mathsf{b}}\oint_{\Gamma_2}\right) \frac{dz_2}{2\pi i\,(\lambda - z_2)} \frac{a(z_2)}{d_{\mathbf{h}}(z_2)} \frac{Q(z_2 - \eta)}{Q(z_2)}\right]^{\epsilon_2 - 1}$$

$$\times \left[\left(\frac{\mathsf{a}}{\mathsf{b}}\oint_{\mathcal{C}_\infty} + \frac{\mathsf{k}_1}{\mathsf{b}}\oint_{\Gamma_1}\right) \frac{dz_1}{2\pi i\,(\lambda - z_1)} \frac{d(z_1)}{d_{\mathbf{h}}(z_1)} \frac{Q(z_1 + \eta)}{Q(z_1)}\right]^{2 - \epsilon_1} \left(\frac{z_1 - z_2}{z_1 - z_2 + \eta}\right)^{(2-\epsilon_1)(\epsilon_2 - 1)}$$

$$\times V(\xi_1^{(h_1)},\ldots,\xi_N^{(h_N)})\, _K\langle \mathbf{h}|, \tag{201}$$

*in which the contour $\mathcal{C}_\infty$ surrounds only the pole at infinity, the contour $\Gamma_2$ surrounds counterclockwise the points $\xi_n$, $1 \leq n \leq N$, and no other poles in the integrand, and the contour $\Gamma_1$*

*surrounds counterclockwise the points $\xi_n - \eta$, $1 \le n \le N$, the point $z_2 - \eta$ if $\epsilon_2 = 1$, and no other poles in the integrand.*

*Similarly, for generic parameters $\lambda_1, \ldots, \lambda_m$, the multiple action of a product of operators $\bar{T}^{(K)}_{\epsilon_2,\epsilon_1}(\lambda_1) \bar{T}^{(K)}_{\epsilon_4,\epsilon_3}(\lambda_2) \ldots \bar{T}^{(K)}_{\epsilon_{2m},\epsilon_{2m-1}}(\lambda_m)$, $\epsilon_i \in \{1,2\}$, $1 \le i \le 2m$, on a generic separate state ${}_K\langle Q|$ of the form (76) can be written as the following sum of contour integrals:*

$$
{}_K\langle Q|\, \bar{T}^{(K)}_{\epsilon_2,\epsilon_1}(\lambda_1)\, \bar{T}^{(K)}_{\epsilon_4,\epsilon_3}(\lambda_2) \ldots \bar{T}^{(K)}_{\epsilon_{2m},\epsilon_{2m-1}}(\lambda_m) = \sum_{\mathbf{h}} \prod_{j=1}^{m} [\mathbf{b}\, d_{\mathbf{h}}(\lambda_j)] \prod_{n=1}^{N} Q(\xi_n^{(h_n)})
$$

$$
\times \prod_{j=m}^{1} \left\{ \left[ \left( \frac{\mathrm{d}}{\mathbf{b}} \oint_{\mathcal{C}_\infty} + \frac{k_2}{\mathbf{b}} \oint_{\Gamma_{2j}} \right) \frac{dz_{2j}}{2\pi i\,(\lambda_j - z_{2j})} \frac{a(z_{2j})}{d_{\mathbf{h}}(z_{2j})} \frac{Q(z_{2j} - \eta)}{Q(z_{2j})} \prod_{k=1}^{j-1} \frac{z_{2j} - \lambda_k - \eta}{z_{2j} - \lambda_k} \right]^{\bar{\epsilon}_{2j}} \right.
$$

$$
\times \left[ \left( \frac{\mathbf{a}}{\mathbf{b}} \oint_{\mathcal{C}_\infty} + \frac{k_1}{\mathbf{b}} \oint_{\Gamma_{2j-1}} \right) \frac{dz_{2j-1}}{2\pi i\,(\lambda_j - z_{2j-1})} \frac{d(z_{2j-1})}{d_{\mathbf{h}}(z_{2j-1})} \frac{Q(z_{2j-1} + \eta)}{Q(z_{2j-1})} \prod_{k=1}^{j-1} \frac{z_{2j-1} - \lambda_k + \eta}{z_{2j-1} - \lambda_k} \right]^{\bar{\epsilon}_{2j-1}} \right\}
$$

$$
\times \prod_{1 \le j < k \le 2m} \left( \frac{z_j - z_k}{z_j - z_k + (-1)^k \eta} \right)^{\bar{\epsilon}_j \bar{\epsilon}_k} V(\xi_1^{(h_1)}, \ldots, \xi_N^{(h_N)})\, {}_K\langle \mathbf{h}|. \quad (202)
$$

*Here we have defined, for $1 \le j \le m$, $\bar{\epsilon}_{2j} = \epsilon_{2j} - 1$ and $\bar{\epsilon}_{2j-1} = 2 - \epsilon_{2j-1}$. The contour $\mathcal{C}_\infty$ surrounds counterclockwise only the pole at infinity, the contours $\Gamma_{2j}$ surround counterclockwise the points $\xi_n$, $1 \le n \le N$, the points $z_{2k-1} + \eta$, $k > j$, and no other poles in the integrand, whereas the contours $\Gamma_{2j-1}$ surround counterclockwise the points $\xi_n - \eta$, $1 \le n \le N$, the points $z_{2k} - \eta$, $k \ge j$, and no other poles in the integrand.*

This result can be proven similarly as Proposition 5.1. It is interesting to note here that the extra contribution in (161)-(162) with respect to (16) and (17) can directly be taken into account by a contribution of the pole at infinity in the multiple integral representations (201) and (202).

Hence, moving the contour as in Proposition 5.2 will simply result in a modification of the weights of the contributions of the different poles, and in particular of the pole at infinity. More precisely, the analog of Proposition 5.2 in the case of the twist $K$ (154) can be formulated as follows:

**Proposition 7.2.** *For generic parameters $\lambda_1, \ldots, \lambda_m$, the multiple action of a product of operators $\bar{T}^{(K)}_{\epsilon_2,\epsilon_1}(\lambda_1) \bar{T}^{(K)}_{\epsilon_4,\epsilon_3}(\lambda_2) \cdots \bar{T}^{(K)}_{\epsilon_{2m},\epsilon_{2m-1}}(\lambda_m)$, $\epsilon_i \in \{1,2\}$, $1 \le i \le 2m$, on a generic separate state ${}_K\langle Q|$ of the form (200) can be written as the following sum of contour integrals:*

$$
{}_K\langle Q|\, \bar{T}^{(K)}_{\epsilon_2,\epsilon_1}(\lambda_1)\, \bar{T}^{(K)}_{\epsilon_4,\epsilon_3}(\lambda_2) \ldots \bar{T}^{(K)}_{\epsilon_{2m},\epsilon_{2m-1}}(\lambda_m) = \sum_{\mathbf{h}} \mathbf{b}^m \prod_{j=1}^{m} d_{\mathbf{h}}(\lambda_j) \prod_{n=1}^{N} Q(\xi_n^{(h_n)})
$$

$$
\times \prod_{j=m}^{1} \left\{ \left[ \left( \frac{k_2 - \mathrm{d}}{\mathbf{b}} \oint_{\mathcal{C}_\infty} + \frac{k_2}{\mathbf{b}} \oint_{\mathcal{C}_j} \right) \frac{dz_{2j}}{2\pi i\,(z_{2j} - \lambda_j)} \frac{a(z_{2j})}{d_{\mathbf{h}}(z_{2j})} \frac{Q(z_{2j} - \eta)}{Q(z_{2j})} \prod_{k=1}^{j-1} \frac{z_{2j} - \lambda_k - \eta}{z_{2j} - \lambda_k} \right]^{\bar{\epsilon}_{2j}} \right.
$$

$$
\times \left[ \left( \frac{k_1 - \mathbf{a}}{\mathbf{b}} \oint_{\mathcal{C}_\infty} + \frac{k_1}{\mathbf{b}} \oint_{\mathcal{C}_j} \right) \frac{dz_{2j-1}}{2\pi i\,(z_{2j-1} - \lambda_j)} \frac{d(z_{2j-1})}{d_{\mathbf{h}}(z_{2j-1})} \frac{Q(z_{2j-1} + \eta)}{Q(z_{2j-1})} \prod_{k=1}^{j-1} \frac{z_{2j-1} - \lambda_k + \eta}{z_{2j-1} - \lambda_k} \right]^{\bar{\epsilon}_{2j-1}} \right\}
$$

$$
\times \prod_{1 \le j < k \le 2m} \left( \frac{z_j - z_k}{z_j - z_k + (-1)^k \eta} \right)^{\bar{\epsilon}_j \bar{\epsilon}_k} V(\xi_1^{(h_1)}, \ldots, \xi_N^{(h_N)})\, {}_K\langle \mathbf{h}|, \quad (203)
$$

where we have defined $\bar{\epsilon}_{2j-1} = 2 - \epsilon_{2j-1}$, $\bar{\epsilon}_{2j} = \epsilon_{2j} - 1$. The contour $\mathcal{C}_\infty$ only surrounds counterclockwise the pole at infinity, whereas the contours $\mathcal{C}_j$, $1 \le j \le 2m$, surround counterclockwise the points $q_n$, $1 \le n \le R$, $\lambda_\ell$, $1 \le \ell \le j$, and no other pole of the integrand.

*Remark* 11. It is interesting to observe that, when $k_1$ and a tend to the same non-zero value $\bar{a}$, whereas $k_2$ and d tend to the same non-zero value $\bar{d}$, the contributions of the pole at infinity become negligible compared to the contributions of the other poles. This is in particular the case when we tend (from non-diagonal values) to a diagonal matrix $K$. The contributions of the pole at infinity also disappear when the matrix $K$ is triangular with c = 0, b $\ne$ 0, a = $k_1$, d = $k_2$.

## 7.3 Correlation functions

We can now compute, for any $2m$-tuple $\epsilon \equiv (\epsilon_1, \ldots, \epsilon_{2m})$, the matrix elements of the form

$$F_m^{(K)}(\tau, \epsilon) = \frac{{}_K\langle Q_\tau | \prod_{j=1}^m E_j^{\epsilon_{2j-1}, \epsilon_{2j}} | Q_\tau \rangle_K}{{}_K\langle Q_\tau | Q_\tau \rangle_K} \tag{204}$$

$$= \frac{{}_K\langle Q_\tau | \bar{T}_{\epsilon_2, \epsilon_1}^{(K)}(\xi_1) \ldots \bar{T}_{\epsilon_{2m}, \epsilon_{2m-1}}^{(K)}(\xi_m) | Q_\tau \rangle_K}{\prod_{k=1}^m \tau(\xi_k) {}_K\langle Q_\tau | Q_\tau \rangle_K}, \tag{205}$$

and their thermodynamic limit

$$\mathcal{F}_m^{(K)}(\epsilon) = \lim_{N\to\infty} \frac{{}_K\langle Q_\tau | \prod_{j=1}^m E_j^{\epsilon_{2j-1}, \epsilon_{2j}} | Q_\tau \rangle_K}{{}_K\langle Q_\tau | Q_\tau \rangle_K}, \tag{206}$$

for $|Q_\tau\rangle_K$ being an eigenstate of the transfer matrix (156) described in the thermodynamic limit by the density of roots $\rho_{\text{tot}}$.

The different steps of the computation follow closely what has been done in the anti-periodic case. We have first to rewrite the multiple integrals in terms of sums on the residues as done in Corollary 5.1. Here, we have just to pay attention to the different weights associated with the residues, i.e. $k_2/b$ or $k_1/b$ for the finite poles and $(k_2 - d)/b$ or $(k_1 - a)/b$ for the poles at infinity. Hence, we can rewrite (204) as a sum over scalar products as in (103). More precisely, the analog of Proposition 5.2 in the case of the twist $K$ (154) can be formulated as follows:

**Proposition 7.2.** *For a given $2m$-tuple $\epsilon \equiv (\epsilon_1, \ldots, \epsilon_{2m}) \in \{1, 2\}^{2m}$, let us define the sets $\alpha_\epsilon^-$ and $\alpha_\epsilon^+$ as in (101)-(102). Then,*

$$F_m^{(K)}(\tau, \epsilon) = \frac{b^{m-s_\epsilon-s'_\epsilon}}{\prod_{k=1}^m \tau(\xi_k)} \sum_{\substack{\bar{\alpha}_\epsilon^- \subset \alpha_\epsilon^- \\ \bar{\alpha}_\epsilon^+ \subset \alpha_\epsilon^+}} (-1)^{(m-\#\bar{\alpha}_\epsilon^- + \#\bar{\alpha}_\epsilon^+)N} (k_1-a)^{s_\epsilon - \#\bar{\alpha}_\epsilon^-}(k_2-d)^{s'_\epsilon - \#\bar{\alpha}_\epsilon^+} \sum_{\{a_j, a'_j\}}$$

$$\times \prod_{j\in\bar{\alpha}_\epsilon^-}\left(\frac{k_1 d(q_{a_j}) \prod_{\substack{k=1 \\ k\in\mathbf{A}_j}}^{R+j-1}(q_{a_j} - q_k + \eta)}{\prod_{\substack{k=1 \\ k\in\mathbf{A}'_j}}^{R+j}(q_{a_j} - q_k)}\right) \prod_{j\in\bar{\alpha}_\epsilon^+}\left(\frac{k_2 a(q_{a'_j}) \prod_{\substack{k=1 \\ k\in\mathbf{A}'_j}}^{R+j-1}(q_k - q_{a'_j} + \eta)}{\prod_{\substack{k=1 \\ k\in\mathbf{A}_{j+1}}}^{R+j}(q_k - q_{a'_j})}\right)$$

$$\times \frac{{}_K\langle \bar{Q}_{\mathbf{A}_{m+1}} | Q_\tau \rangle_K}{{}_K\langle Q_\tau | Q_\tau \rangle_K}. \tag{207}$$

*In (207), the first summation is taken over all subsets $\bar{\alpha}_\epsilon^-$ of $\alpha_\epsilon^-$ and $\bar{\alpha}_\epsilon^+$ of $\alpha_\epsilon^+$, whereas the second summation is taken over the indices $a_j$ for $j \in \bar{\alpha}_\epsilon^-$ and $a'_j$ for $j \in \bar{\alpha}_\epsilon^+$ defined as in (104)-(106). Moreover, $\bar{Q}_{\mathbf{A}_{m+1}}$ is the polynomial of degree $\bar{R} = \#\mathbf{A}_{m+1} = R + m - \#\bar{\alpha}_- - \#\bar{\alpha}_+$ defined in terms of the roots $q_1, \ldots, q_R$ of $Q_\tau$ and of $q_{R+j} \equiv \xi_j$, $1 \le j \le m$, as in (107).*

The corresponding scalar products are then computed using the identities of section 3.2 of [113]: in [113], these scalar products were shown to admit a Slavnov's type determinant formula (see formula (3.46) of [113]) associated to a twist parameter $\mu$, which in our current case reads $\mu = k_2/k_1$. More precisely, in the case $\bar{R} \geq R$, i.e. $m \geq \#\bar{\alpha}_- + \#\bar{\alpha}_+$, we have

$$\frac{{}_K\langle \bar{Q}_{\mathbf{A}_{m+1}} | Q_\tau \rangle_K}{{}_K\langle Q_\tau | Q_\tau \rangle_K} = (-1)^{N(\bar{R}-R)}(1-\mu)^{R-\bar{R}} \frac{\prod_{j=1}^{\bar{R}}\left[\mu\, a(\bar{q}_j) \prod_{k=1}^R (q_k - \bar{q}_j + \eta)\right]}{\prod_{j=1}^R\left[\mu\, a(q_j) \prod_{k=1}^R (q_k - q_j + \eta)\right]}$$
$$\times \frac{V(q_R, \ldots, q_1)}{V(\bar{q}_{\bar{R}}, \ldots, \bar{q}_1)} \frac{\det_{\bar{R}} \mathcal{M}^{(\mu)}(\mathbf{q}|\bar{\mathbf{q}})}{\det_R \mathcal{N}^{(\mu)}(\mathbf{q})}, \quad (208)$$

with

$$\left[\mathcal{M}^{(\mu)}(\mathbf{q}|\bar{\mathbf{q}})\right]_{j,k} = \begin{cases} t(q_j - \bar{q}_k) - \mu^{-1}\mathfrak{a}_Q(\bar{q}_k)\, t(\bar{q}_k - q_j) & \text{if } j \leq R, \\ (\bar{q}_k)^{j-R-1} - \mu^{-1}\mathfrak{a}_Q(\bar{q}_k)(\bar{q}_k + \eta)^{j-R-1} & \text{if } j > R, \end{cases} \quad (209)$$

$$\left[\mathcal{N}^{(\mu)}(\mathbf{q})\right]_{j,k} = \left[\mathcal{M}^{(\mu)}(\mathbf{q}|\mathbf{q})\right]_{j,k} = -\mu^{-1}\mathfrak{a}'_Q(q_j)\delta_{j,k} + K(q_j - q_k),$$
$$= \frac{\mathfrak{a}'_Q(q_j)}{\mathfrak{a}_Q(q_j)}\delta_{j,k} + K(q_j - q_k), \quad (210)$$

in which we have used the notations (111)-(112) and (115), and the Bethe equations

$$-\mu^{-1}\mathfrak{a}_Q(q_j) = 1, \qquad j = 1, \ldots, R. \quad (211)$$

Note that, for $\{\bar{q}_1, \ldots, \bar{q}_{\bar{R}}\} \subset \{q_1, \ldots, q_R\} \cup \{\xi_1, \ldots, \xi_m\}$, the matrices (209) and (210) coincide respectively with (113) and (114), i.e. the explicit $\mu$-dependence disappears.

It now remains to identify the terms in the sum (207) which are vanishing and non-vanishing in the thermodynamic limit. Since the the matrices (209) and (210) coincide respectively with (113) and (114), the direct analog of Proposition 6.1 for the ratio of scalar products in the $K$-twisted case still holds. In particular, the results (119) and (120) remain valid in this case.

Let us observe that, for a given $2m$-tuple $\boldsymbol{\epsilon} \equiv (\epsilon_1, \ldots, \epsilon_{2m})$,

$$s_\epsilon + s'_\epsilon = m_A + m_D + 2m_C, \quad (212)$$

where $s_\epsilon$ and $s'_\epsilon$ are defined as in (101)-(102) and $m_X$ is the number of $X^{(K)}(\lambda)$ in (205), for $X \in \{A, B, C, D\}$, and let us first consider the case

$$s_\epsilon + s'_\epsilon \leq m, \qquad \text{i.e.} \qquad m_C \leq m_B. \quad (213)$$

We can then repeat the first part of the proof of Corollary 6.1 and derive that the only non-vanishing elementary blocks under the condition (213) are those for which $s_\epsilon + s'_\epsilon = m$, there the only contributing terms are the ones for which the pole at infinity does not contribute, i.e. for which $\bar{\alpha}^\pm_\epsilon = \alpha^\pm_\epsilon$. If now

$$m < s_\epsilon + s'_\epsilon, \qquad \text{i.e.} \qquad m_B < m_C, \quad (214)$$

and if moreover $\mathsf{c} \neq 0$, we can repeat all the computations in the SoV basis given by the eigenbasis of $C^{(K)}(\lambda)$. All the steps that we have described here repeat in the same way but the role of $B$ and $C$ are exchanged. It is easy to verify than the elementary blocks vanish under the condition (214) since, with this construction, the number of sums that we generate is smaller than the order with which the scalar products go to zero in the thermodynamic

limit, as explained in the first part of the proof of Corollary 6.1. If instead $\mathsf{c} = 0$, i.e. if the matrix $K$ is triangular, then the contributions of the poles at infinity disappear (see Remark 11), so that the first sum in (207) reduces to the term $\bar{\alpha}_\epsilon^+ = \alpha_\epsilon^+$ and $\bar{\alpha}_\epsilon^- = \alpha_\epsilon^-$. We can then conclude from the fact that the scalar product is vanishing when $\deg \bar{Q}_{\mathbf{A}_{m+1}} < \deg Q_\tau$, with here $\deg \bar{Q}_{\mathbf{A}_{m+1}} = \deg Q_\tau + m - s_\epsilon - s_\epsilon'$, see (119)[15], so that the corresponding elementary blocks also vanish (even in finite size) under the condition (214).

Therefore, we have proven that

$$
\mathcal{F}_m^{(K)}(\epsilon) = \delta_{s_\epsilon + s_\epsilon', m} \lim_{N \to \infty} \prod_{k=1}^m \frac{1}{\tau(\xi_k)} \sum_{\{a_j, a_j'\}} \prod_{j \in \alpha_\epsilon^-} \left( \frac{\mathsf{k}_1 \, d(q_{a_j}) \prod_{\substack{k=1 \\ k \in \mathbf{A}_j}}^{R+j-1} (q_{a_j} - q_k + \eta)}{\prod_{\substack{k=1 \\ k \in \mathbf{A}_j'}}^{R+j} (q_{a_j} - q_k)} \right)
$$

$$
\times \prod_{j \in \tilde{\alpha}_\epsilon^+} \left( \frac{\mathsf{k}_2 \, a(q_{a_j'}) \prod_{\substack{k=1 \\ k \in \mathbf{A}_j'}}^{R+j-1} (q_k - q_{a_j'} + \eta)}{\prod_{\substack{k=1 \\ k \in \mathbf{A}_{j+1}}}^{R+j} (q_k - q_{a_j'})} \right) \frac{{}_K\langle \bar{Q}_{\mathbf{A}_{m+1}} | Q_\tau \rangle_K}{{}_K\langle Q_\tau | Q_\tau \rangle_K}, \quad (215)
$$

where, as already mentioned, the ratio of scalar products is computed in the thermodynamic limit by the formula (120) of Proposition 6.1. We have to use now the analog of formulas (144) and (145) in the $K$-twisted case, i.e.

$$
\tau(\xi_k) = \mathsf{k}_2 \, a(\xi_k) \frac{Q_\tau(\xi_k - \eta)}{Q_\tau(\xi_k)}, \qquad k = 1, \ldots, N, \tag{216}
$$

$$
\mathsf{k}_1 \, d(q_{a_j}) = -\mathsf{k}_2 \, a(q_{a_j}) \frac{Q_\tau(q_{a_j} - \eta)}{Q_\tau(q_{a_j} + \eta)}, \qquad \forall \, a_j \le R, \tag{217}
$$

and the observation that $d(q_{a_j}) = 0$ for any $a_j > R$, to rewrite the non-zero terms of (215) in a form that coincides with (146)-(147).

Hence, we have shown the following result:

**Proposition 7.3.** *For any $2m$-tuple $\epsilon \equiv (\epsilon_1, \ldots, \epsilon_{2m})$, the matrix element of the form (206) in the $K$-twisted chain coincides, in the thermodynamic limit, with its counterpart (148) in the anti-periodic or periodic chain, i.e.*

$$
\mathcal{F}_m^{(K)}(\epsilon) = \mathcal{F}_m(\epsilon), \tag{218}
$$

*and is given by the multiple integral representations (149), (153). In particular, this matrix elements vanishes when $s_\epsilon + s_\epsilon' \neq m$, where $s_\epsilon$, $s_\epsilon'$ are defined as in (101)-(102).*

In other worlds, we have here explicitly shown that — as expected from physical arguments — the ground state correlation functions of the XXX spin 1/2 chain with quasi-periodic boundary conditions do not depend, in the thermodynamic limit, on the particular boundary condition we consider, i.e. on the particular form of the twist matrix $K$, and coincide with the correlation functions of the periodic chain in the thermodynamic limit, at least for non-diagonal twists. Of course, the same statement can be proven for diagonal twist, by developing the same computations in the algebraic Bethe Ansatz framework as done in the periodic case in [19].

---

[15]This can also be seen from the fact that, in the triangular case, ${}_K\langle \bar{Q}_{\mathbf{A}_{m+1}} | \in \mathcal{H}_{-N/2, \ldots, \bar{R} - N/2}^*$ with $\deg \mathbf{A}_{m+1} = \bar{R}$ whereas $| Q_\tau \rangle_K \in \mathcal{H}_{R-N/2, \ldots, N/2}$ with $\deg Q_\tau = R$ (see Remark 10).

# 8 Conclusion

In this paper, we have explained how to compute correlation functions in the quantum SoV framework, and shown that it is possible to obtain, in this framework, the same kind of results as in the algebraic Bethe Ansatz framework [19] or the $q$-vertex operator approach [24]. To this aim, we have considered a very simple example, the twisted XXX spin chain.

One of the difficulties of the SoV approach for its applicability to physical systems comes from the fact that all results are a priori obtained in terms of the non-physical inhomogeneity parameters that have to be introduced for the method to work. Getting rid of these inhomogeneity parameters, i.e. taking the homogeneous limit, may be a very non-trivial task: at the level of the spectrum, we naturally obtain a description in terms of a discrete Baxter TQ-equation that we need to reformulate into a more conventional one [112]; the determinant representations for the scalar products that we naturally obtain also need to be transformed into more tractable expressions [113]; finally, the action of local operators on separate states involves the inhomogeneity parameters in a very intricate way, and needs to be reformulated.

This last point is crucial if we want to use this approach for the direct computation of correlation functions, and bring the SoV approach to same level of achievement as the algebraic Bethe Ansatz [19] or the $q$-vertex operator approach [24]. In this paper, we have therefore explained how to transform the SoV action into a more conventional one, involving the roots of the Baxter Q-function (the "Bethe roots") rather than the inhomogeneity parameters. More precisely, we have expressed these actions using multiple contour integrals: taking the residues inside the contours, we recover the SoV action in terms of the inhomogeneity parameters; taking the residues outside the contours, we obtain an ABA-type action, in terms of "Bethe roots". Note that, doing this, we also obtain some extra contributions from the pole at infinity. In fact, the correlation functions of the (non-diagonally) twisted XXX chain in finite volume involve many additional contributions with respect to the periodic or diagonally-twisted one, since the spin $S^z$ is no longer conserved. We have explicitly shown here that all these extra contributions are vanishing in the thermodynamic limit, hence leading in this limit to the same result as in the periodic case.

We expect our approach to correlation functions in SoV to be generalizable to more complicated models. A natural question in this respect concerns the (anti-periodic) XXZ chain which, contrary to what happens in the periodic case with Bethe Ansatz, is not a trivial generalization of the XXX case: new difficulties appear due to the fact that the Baxter Q-function is no longer a usual trigonometric polynomial [112, 131]. We intend to solve this problem in a future publication (see [135] for first results on scalar products and form factors). Another natural and interesting question concerns the correlation functions of open chains (XXX or XXZ) with non-diagonal boundaries, for which preliminary results have already been obtained concerning scalar products of separate states [115, 134].

The computation of correlation functions for quantum integrable models associated to the higher rank cases is then a next natural target of great relevance. Given the recent notable achievements [6, 88, 117–129] in their SoV description, with first results available also on compact scalar product formulae, the task to compute correlation functions in higher rank seems very promising nowadays. This is, in particular, the case if one can compute the action of the local operators in the simplified SoV basis introduced in [127] for the rank 2 models. Indeed, under this choice the SoV measure simplify considerably and rank 1 Slavnov's type determinants appear for the scalar product of separate states with transfer matrix eigenvectors. Then, the manipulations described in the present article for these determinants in the thermodynamic limit should be extendible to the higher rank case, under the assumption of a suitable description of the density of the Bethe roots in this limit. Nevertheless, one should point out that at the current stage the main difficulty left is exactly the computation of the action of

local operators on these higher rank SoV basis. This represents a new problem mainly for the complexity of the action of the monodromy matrix elements on the SoV basis. In fact, one should recall that up now only the action of the transfer matrix has been computed in these SoV basis and this was possible thanks to the use of the fusion relations satisfied by the same transfer matrices. How to generalize this to the other elements of the monodromy matrix is currently under analysis but still at an early stage to be here discussed.

## Acknowledgements

G.N. would like to thank J.M. Maillet for his interest and for several stimulating discussions on the present paper and related subjects. G.N. is supported by CNRS and ENS de Lyon. V.T. is supported by CNRS.

## A  On elementary blocks for similar transfer matrices

In this paper, as in [19, 24], we have not computed the more general correlation functions but their elementary buildings blocks, i.e. quantities of the form (70). Here we make some comments on the role of the $GL(2)$ transformations on such elementary building blocks for the quasi-periodic XXX chains, and on the consequences for the computation of such elementary building blocks for similar transfer matrices.

Due to the $GL(2)$-invariance of the XXX R-matrix (4), the transfer matrix of the periodic chain,

$$\mathcal{T}^{(I)}(\lambda) = \text{tr}_0 T_0(\lambda), \tag{219}$$

satisfies, for any invertible matrix $\gamma \in GL(2)$, the invariance property

$$[\mathcal{T}^{(I)}(\lambda), \Gamma] = 0 \qquad \Gamma = \bigotimes_{n=1}^{N} \gamma_n. \tag{220}$$

As a consequence of this invariance, we obtain the following identity on the elementary blocks of correlation functions:

$$\frac{\langle \Psi_I | \prod_{j=1}^{m} E_j^{\epsilon_{2j-1}, \epsilon_{2j}} | \Psi_I \rangle}{\langle \Psi_I | \Psi_I \rangle} = \frac{\langle \Psi_I | \Gamma \left( \prod_{j=1}^{m} E_j^{\epsilon_{2j-1}, \epsilon_{2j}} \right) \Gamma^{-1} | \Psi_I \rangle}{\langle \Psi_I | \Psi_I \rangle}, \tag{221}$$

where $|\Psi_I\rangle$ denotes any eigenstate of (219).

However, one should point out that, as soon as we consider quasi-periodic boundary conditions with a non-identity twist $K$, the $GL(2)$ invariance of the transfer matrix is lost. Hence, in general, for such a twisted chain in finite volume,

$$\frac{\langle \Psi_K | \prod_{j=1}^{m} E_j^{\epsilon_{2j-1}, \epsilon_{2j}} | \Psi_K \rangle}{\langle \Psi_K | \Psi_K \rangle} \neq \frac{\langle \Psi_K | \Gamma \left( \prod_{j=1}^{m} E_j^{\epsilon_{2j-1}, \epsilon_{2j}} \right) \Gamma^{-1} | \Psi_K \rangle}{\langle \Psi_K | \Psi_K \rangle}, \tag{222}$$

for any $\gamma$ which does not commute with $K$, where $|\Psi_K\rangle$ denotes a given eigenstate of the $K$-twisted transfer matrix $\mathcal{T}^{(K)}(\lambda)$. Let use now consider the twist $K_\gamma = \gamma^{-1} K \gamma$. The $K_\gamma$-twisted transfer matrix is then given by

$$\mathcal{T}^{(K_\gamma)}(\lambda) = \Gamma^{-1} \mathcal{T}^{(K)}(\lambda) \Gamma, \tag{223}$$

and admits the following eigenstates:

$$\langle \Psi_{K_\gamma} | = \langle \Psi_K | \Gamma, \qquad |\Psi_{K_\gamma}\rangle = \Gamma^{-1} |\Psi_K\rangle. \tag{224}$$

Hence,

$$\frac{\langle \Psi_K | \Gamma \left( \prod_{j=1}^m E_j^{\epsilon_{2j-1},\epsilon_{2j}} \right) \Gamma^{-1} | \Psi_K \rangle}{\langle \Psi_K | \Psi_K \rangle} = \frac{\langle \Psi_{K_\gamma} | \prod_{j=1}^m E_j^{\epsilon_{2j-1},\epsilon_{2j}} | \Psi_{K_\gamma} \rangle}{\langle \Psi_{K_\gamma} | \Psi_{K_\gamma} \rangle}, \tag{225}$$

and the inequality (222) can be equivalently rewritten as

$$\frac{\langle \Psi_K | \prod_{j=1}^m E_j^{\epsilon_{2j-1},\epsilon_{2j}} | \Psi_K \rangle}{\langle \Psi_K | \Psi_K \rangle} \neq \frac{\langle \Psi_{K_\gamma} | \prod_{j=1}^m E_j^{\epsilon_{2j-1},\epsilon_{2j}} | \Psi_{K_\gamma} \rangle}{\langle \Psi_{K_\gamma} | \Psi_{K_\gamma} \rangle}, \tag{226}$$

for any $\gamma$ which does not commute with $K$. That is, the same elementary block associated to two similar transfer matrices (or equivalently to two similar twist matrices) do not in general coincide in the finite chain. Of course, the equality may be recovered in the thermodynamic limit, and we have indeed shown in section 7 that

$$\mathcal{F}_m^{(K)}(\epsilon) = \mathcal{F}_m^{(K_\gamma)}(\epsilon), \tag{227}$$

provided $|\Psi_K\rangle$ and $|\Psi_{K_\gamma}\rangle$ are described in the thermodynamic limit by the density of Bethe roots (57) on the real axis.

Finally, we want to point out that it is a priori not easy to deduce the expression of an elementary block in the gauge transform model from the ones that we can compute in the original model. In this respect, the exact computations of the elementary blocks that we have developed in the SoV framework for the quasi-periodic boundary conditions associated to non-diagonal twist matrices $K$ is an interesting set of results in their own and not only for their ability to describe our SoV approach to correlation functions.

In fact, let $K$ be non-diagonal but diagonalizable and $K_\gamma$ diagonal, then the similarity relations (223) may suggest that, in the XXX chain, one can compute the elementary blocks of the quasi-periodic boundary condition associated to the twist $K$ in terms of those of the twist $K_\gamma$, as it follows:

$$\frac{\langle \Psi_K | \prod_{j=1}^m E_j^{\epsilon_{2j-1},\epsilon_{2j}} | \Psi_K \rangle}{\langle \Psi_K | \Psi_K \rangle} = \frac{\langle \Psi_{K_\gamma} | \left[ \Gamma^{-1} \left( \prod_{j=1}^m E_j^{\epsilon_{2j-1},\epsilon_{2j}} \right) \Gamma \right] | \Psi_{K_\gamma} \rangle}{\langle \Psi_{K_\gamma} | \Psi_{K_\gamma} \rangle}. \tag{228}$$

The main problem with this approach is that the matrix element on the r.h.s. of the above identity is not one simple elementary block but in general the sum of $4^m$ different elementary blocks. Now by the symmetry satisfied by $\mathcal{T}^{(K_\gamma)}(\lambda)$ some of them can be proven to be zero and we know how to compute all the others in the ABA framework but we have still to sum all them up to get just one elementary block associated to the transfer matrix $\mathcal{T}^{(K)}(\lambda)$. Our previous discussion tells us that this sum has to reproduce in the thermodynamic limit always the same elementary block. We have proven it by our direct SoV approach, however to prove it only in the ABA framework seems a complicate task as it is equivalent to prove that the large sum of nonzero elementary blocks obtained expanding the difference:

$$\frac{\langle \Psi_{K_\gamma} | \left[ \Gamma^{-1} \left( \prod_{j=1}^m E_j^{\epsilon_{2j-1},\epsilon_{2j}} \right) \Gamma \right] | \Psi_{K_\gamma} \rangle}{\langle \Psi_{K_\gamma} | \Psi_{K_\gamma} \rangle} - \frac{\langle \Psi_{K_\gamma} | \prod_{j=1}^m E_j^{\epsilon_{2j-1},\epsilon_{2j}} | \Psi_{K_\gamma} \rangle}{\langle \Psi_{K_\gamma} | \Psi_{K_\gamma} \rangle}, \tag{229}$$

has to be zero in the thermodynamic limit.

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
