# Peer review of "Correlation functions by Separation of Variables: the XXX spin chain"

_SciPost Physics, doi:SciPost Phys. 10, 006 (2021)_

## Round 2 · Referee Report · Anonymous (Referee 1) · 2020-10-1

Strengths

1. Explicit implementation of the SoV approach in the thermodynamic limit.

2 - Proof that correlation functions in the thermodynamic limit do not depend on the specific choice of boundary twist.

3 - Rewriting of the action of monodromy elements on separated states in terms of Bethe roots instead of inhomogeneities.

Weaknesses

1 - Restricted to gl(2)-type models using methods which are somewhat outdated and not well suited to higher-rank generalisations.

Report

The authors have considered the problem of computing correlation functions in the thermodynamic limit for the XXX gl(2) spin chain. They begin by reviewing the separation of variables procedure for this spin chain with anti-periodic boundary conditions and construct the appropriate separated variable states and compute the action of the monodromy matrix elements on these states. Then, they show how this action can be rewritten in terms of a family of contour integrals which can then be re-expressed as a sum over Bethe roots, which is one of the central results of the paper. Next, the authors combine their results to compute zero-temperature correlation functions. Finally the authors generalise their computations to the case of more general non-diagonal twists and nicely show that the correlation functions in the thermodynamic limit are independent of the specific choice of twist.

Overall I am impressed with the quality of the paper and it certainly merits publication, but I have a few changes which I ask the authors to consider implementing.

Requested changes

1 - The most important change which I request the authors implement is to add at least some discussion about possible generalisations to higher-rank. Given the recent tremendous progress in developing the separation of variables procedure for higher-rank I feel that such a discussion is crucial for the completeness of the paper. To elaborate further, here the situation is a bit simpler because the action of all A,B,C,D on the SoV states can be easily computed, since one acts diagonally, two act as simple raising / lowering operators and the remaining one can be computed from the quantum determinant. This is no longer true at higher rank where in principle the only thing one knows is the action of the transfer matricies and the associated fusion relations. Is this enough to reconsruct the action of all monodromy elements?

I should stress that I do not expect the authors to do any actual computations / derivations / proofs relating to this, but to at least comment on these potential difficulties and how they might be overcome, for example in the Conclusion section of the paper where the authors already discuss some possible generalisations to other more complicated models.

2 - Under 3.20 please define what a “generalised Bethe state” is and perhaps also explain how it is different from a usual Bethe state (which I would consider to be any eigenvector of the transfer matrix).

3 - Finally, here are some typos / grammatical errors I noticed:

“Here develop” -> “develop here”
“Like here poles” -> like poles
“In related SoV framework” -> “in a related SoV framework” or perhaps just “in the SoV framework”.
“Implement here our approach” -> “implement our approach here”
“Not necessarily solution” -> “not necessarily a solution”
“Operators zeros” -> operator zeros”
“Results than in” -> “results as in”
“One of the difficulty” -> “one of the difficulties”
“Achievement than the” -> “achievement as the”

---

## Round 2 · Referee Report · Anonymous (Referee 2) · 2020-10-12

Strengths

1 - Reformulation of the SoV correlation functions in term of the Bethe roots

Weaknesses

1 - Original References not always accurate

2- No discussion of the problematic of the TQ equation with an additional term to reconstruct correlation function in term of the Bethe roots.

Report

The authors study the correlation function of the twisted XXX spin 1/2 chain by means of the separation of variables (SoV). They recover
known results about the model. Moreover they show the independence of the model to the twist in the thermodynamical limit.

The paper is of interest and deserve publication. But some points must be improved before this.

Requested changes

- p3 the authors could itemize the essential ingredients to study correlation function. The Slavnov formula’s can be obtain by means of the algebraic Bethe ansatz for models without U(1) symmetry (arXiv:1506.06550, arXiv:1507.03242, arXiv:1805.11323, arXiv:1906.06897, arXiv:1908.00032,arXiv:2005.11224 ). The lack of reference state is due to the lack of U(1) symmetry of the Hamiltonian, but Bethe ansatz (off diagonal or modified algebraic) still work in that case.

- p4 the authors must discuss more clearly the situation about TQ equation with an additional term (originally from Off diagonal Bethe ansatz approach) and the case with non polynomial Q function. They can gives more details about what is a Slavnov type determinant.

- p7 the authors could gives a explicit formula for the basis |h>.

- p8-9 Did the author can find the action of the transfert matrix on an arbitrary separate state ?

- p13 typo eq (4.16) K->k.

- p14 Did the author have more reference about the ground state of the diagonal twist case ?

- p16 ref [98] the original reference must be added

---

## Round 3 · Referee Report · Anonymous (Referee 1) · 2020-12-9

Report

I am satisfied with the changes the authors have implemented, in particular for their nice discussion regarding the possible generalisation of their techniques to higher rank models and for clearing up some potentially confusing terminology used in the paper.

I am now happy to recommend the paper for publication.

---

## Round 3 · Author Response

Dear Editor,

We would like to thank the referees for their attentive readings, clarification requests and for pointing out some further existing literature. We have implemented some modifications on the text of our manuscript to take them into account, mainly in the introduction and the conclusion. Let us answer separately to the two referees as it follows.

Answer to Referee 1: 1. We agree with the referee about the relevance of the extension of our SoV method to compute correlation functions to more advanced models as those associated to the higher rank cases. We have added in our Conclusion some direct comments on this and on the existing literature which defines the basis for such an extension. The referee points out one of the central issues that has to be solved in the higher rank case in order to extend our SoV approach to correlation functions, i.e. how to compute the action of local operators on transfer matrix eigenstates in SoV framework. Thanks to the reconstruction formula [18] this is mainly equivalent to the computation of the action of the monodromy matrix elements on the SoV basis. In [118], the fusion equations are indeed the main ingredient to compute the action of the transfer matrix on a separate state and to fix its form in order to be a transfer matrix eigenvector. It is then natural to expect that the fusion of the monodromy matrix themselves should play a role in this “out of diagonal” type of action computations. Some ideas are currently under investigation in this direction but they are still too premature to comment openly on them. Another important point is the derivation of efficient scalar product formulae of determinant type. The results in [127] are going in this direction, see also [128-129]. In particular, we would like to mention that in [127], a special choice of the convector/vector SoV bases has been defined which manifests a strong analogy in the SoV measure to the one for the rank one case and leads to Slavnov’s type determinant formulae for scalar product of separate states with transfer matrix eigenstates. Such results when accompanied by the computation of the action of local operators are at the basis of the generalization of our SoV method toward correlation functions. Said that, we want to stress that the main aim of the present manuscript is to introduce and illustrate, for a simple set of models, a very much needed method to compute correlation functions in the SoV framework. To our knowledge suck a method is not outdated but rather a complete novelty in the literature and hopefully of potential large use. Moreover, while computing correlation functions for higher rank model would surely be an outstanding result, this is not the next natural subject of our research. Indeed, we want to stress that there is a large variety of rank one models whose correlation function are not available in the literature and whose computations would represent a fundamental achievement. This is for example the case of the open XXX/XXZ/XYZ spin chain with general boundary conditions.
2. We agree with the referee suggestion and we have added some text to explain it. In fact, we call them in that way because they have a Bethe Ansatz form but they are not derived by Bethe Ansatz. Indeed, no ansatz is done on them and they are just a simple rewriting of the SoV separate state description of transfer matrix eigenvectors. 3. We have implemented the text suggestion of the referee.

Answer to Referee 2: We agree with the referee that our introduction missed the discussion of some relevant literature and we have modified accordingly our manuscript. 1. We have listed the essential ingredients for computation of correlation functions as required by the referee. This has given us the opportunity to further stress the role of the knowledge of the density of the Bethe roots of the ground state to compute correlation functions, as first remarked in [19]. Concerning scalar products for models without U(1) symmetries, we agree in the interest of the developments in the framework of the so-called modified ABA (MABA), and we have reported them accordingly to the referee suggestions. Concerning the interesting paper 2005.11224, let us comment that it is subsequent to our manuscript, nevertheless we have decided to take it into account. 2. At page 4-5, we have recalled the existing approaching by inhomogeneous TQ Baxter equations and the non-polynomial Q-operator with the associated literature. 3. Let us comment that we didn’t write explicitly the form of the SoV basis as it does not play any role in the computation of the correlation functions. As it is known now by the paper [113], we can have different representations for the same states and currently in the text we are explicitly referring to this paper for these different explicit forms. 4. The question of the referee is not completely clear to us. We can compute the action of the transfer matrix on a generic separate state. This can be done by using the SoV representation of the Yang-Baxter algebra in the SoV basis as given in page 8. It is exactly this type of computation which fix uniquely and without any ansatz the form of the transfer matrix eigenstate as given in pages 9-10, which are special instances of separate states. We do not recall these steps here as they are well known results in the literature, see for example [117] for a recall in the XXX case. 5. Thanks for the remarked typo, we correct it. 6. We are not at the knowledge of any further reference. 7. We have added the reference to the paper where this homogeneous TQ equation first appeared and clarified the role played in [110] in proving it.

Finally, from the referee reports, we would like to make some remarks. a) While not being the main aim of our manuscripts, the results here derived are to our knowledge entirely new. They were only expected on physical basis but not proven elsewhere in the literature by any other method. b) The scalar products of Slavnov form first appeared and have been proven for twisted XXX chains in the SoV framework in our previous paper [113]. The scalar products in the MABA framework have a similar form to those derived in the SoV framework (i.e. there they are rewritten as the determinant of the Jacobian of the transfer matrix eigenvalues or some local deformations of it). Nevertheless, it seems that in MABA framework the description of the transfer matrix eigenvalues are always given in terms of inhomogeneous TQ-equations. This is not always the case in the SoV framework. On the one hand, this makes complicate a direct connection between these formulae. On the other hand, to our knowledge, the analytic identification of the ground state at the thermodynamic limit is a very much complicate and still unsolved task for the inhomogeneous TQ equation (at least in terms of a density function solution of an integral equation in the thermodynamic limit with some control on the finite-size corrections, as from usual Bethe equations). This is one of the fundamental ingredients to analytically compute correlation functions, which is still missing in the literature for this inhomogeneous case.
Nevertheless, when considering the XXX spin chains with general quasi-periodic boundary conditions in the SoV framework, we don’t need to use such an inhomogeneous TQ equation (we could of course, but it would be counter productive for our purpose), since it is easy to rewrite the SoV spectrum and eigenstates in terms of solutions of a usual (homogeneous) TQ equation, as done in [113]. From which we can easily obtain a simple description of the ground state in the thermodynamic limit in terms of a density function and so implement as described in the manuscript the computation of correlation functions.

---

## Round 3 · List of Changes

See the author comments which list the changes made according to each referee's suggestions

---

## Editorial Decision

published